# Coalescence of bacterial groups originating from urban runoffs and artificial infiltration systems among aquifer microbiomes

Yannick Colin[1‡], Rayan Bouchali[1], Laurence Marjolet[1], Romain Marti[1], Florian Vautrin[1,2], Jérémy Voisin[1,2], Emilie Bourgeois[1], Veronica Rodriguez-Nava[1], Didier Blaha[1], Thierry Winiarski[2], Florian Mermillod-Blondin[2] and Benoit Cournoyer[1]

[1]University of Lyon, UMR Ecologie Microbienne Lyon (LEM), Research Team "Bacterial Opportunistic Pathogens and Environment", Université Lyon 1, CNRS 5557, INRA 1418, VetAgro Sup, 69680 Marcy L'Etoile, France.

[2]University of Lyon, UMR Laboratoire d'Ecologie des Hydrosystèmes Naturels et Anthropisés (LEHNA), Université Lyon 1, CNRS 5023, ENTPE, 69622 Villeurbanne, France.

[‡] Present address : Normandie Université, UNIROUEN, UNICAEN, UMR CNRS 6143, Morphodynamique Continentale et Côtière, 76000 Rouen, France

**Running title: Y. Colin et al.: Urban runoff bacteria among recharged aquifer**

**Keywords:** Stormwater infiltration; Microbial contamination; Aquifer; Source-tracking; Biofilms

**Correspondence :** yannick.colin@univ-rouen.fr / benoit.cournoyer@vetagro-sup.fr

**Abstract.** The invasion of aquifer microbial communities by aboveground microorganisms, a phenomenon known as community coalescence, is likely to be exacerbated in groundwaters fed by stormwater infiltration systems (SIS). Here, the incidence of this increased connectivity with upslope soils and impermeabilized surfaces was assessed through a meta-analysis of 16S rRNA gene libraries. Specifically, 16S rRNA V5-V6 DNA sequences from free-living and attached aquifer bacteria (i.e., water and biofilm samples) were analysed upstream and downstream a SIS, and compared with those from bacterial communities from watershed runoffs, and surface sediments from the SIS detention and infiltration basins. Significant bacterial transfers were inferred by the SourceTracker Bayesian approach, with 23 to 57% of the aquifer bacterial biofilms being composed of taxa from aboveground sediments and urban runoffs. Sediments from the detention basin were found more significant contributors of taxa involved in the build-up of these biofilms than soils from the infiltration basin. Inferred taxa among the coalesced biofilm community were predicted to be high in hydrocarbon degraders such as *Sphingobium* and *Nocardia.* The 16S rRNA-based bacterial community structure of the downstream SIS aquifer waters showed lower coalescence with aboveground taxa (8 to 38%) than those of biofilms, and higher numbers of taxa predicted to be involved in the N- and S-cycles. A DNA marker named *tpm* enabled the tracking of bacterial species from 24 genera including the *Pseudomonas*, *Aeromonas* and *Xanthomonas*, among these communities. Several *tpm* sequence types were found to be shared between the aboveground and aquifer samples. Reads related to *Pseudomonas* were allocated to 50 species, of which 16 were found in the aquifer samples. Several of these aquifer species were found to be involved in denitrification but also hydrocarbon degradation (*P. aeruginosa*, *P. putida*, and *P. fluorescens*). Some *tpm* sequence types allocated to *P. umsongensis* and *P. chengduensis* were found to be enriched among the *tpm*-harboring bacteria, respectively of the aquifer biofilms, and waters. Reads related to *Aeromonas* were allocated to 11 species, but only those from *A. caviae* were recovered aboveground and in the aquifer samples. Some *tpm* sequence types of the *X. axonopodis* phytopathogen were recorded in higher proportions among the *tpm*-harboring bacteria of the aquifer waters than in the aboveground samples. A significant coalescence of microbial communities from an urban watershed with those of an aquifer was thus observed, and recent aquifer biofilms were found to be significantly colonized by runoff opportunistic taxa able to use urban C-sources from aboveground compartments.

## 1 Introduction

Urbanization exerts multiple pressures on natural habitats and particularly on aquatic environments (Konrad and Booth, 2005; McGrane, 2016; Mejía and Moglen, 2009). The densification of urban areas, combined with the conversion of agricultural and natural lands into urban land-use, led to the replacement of vegetation and open fields by impervious urban structures (*i.e.*, roads, rooftops, side-walks and parking lots) (Barnes et al., 2001). These impervious structures reduce the infiltration capacity of soils. They also exacerbate the speed and volume of stormwater runoff that favor soil erosion, flooding events, and affect adversely natural groundwater recharge processes (Booth, 1991; Shuster et al., 2005). Due to these consequences, stormwater infiltration systems (SIS) or managed aquifer recharged systems (MAR) have been developed over the last decades, and are gaining more interest in developed countries (Pitt et al., 1999). Such practices reduce direct stormwater discharges to surface waters and alleviate water shortages (Barba et al., 2019; Dillon et al., 2008; Marsalek and Chocat, 2002). However, stormwater represents a major source of nonpoint pollution, and its infiltration into the ground may have adverse ecological and sanitary impacts (Chong et al., 2013; Pitt et al., 1999; Vezzaro and Mikkelsen, 2012).

The vadose zone of a SIS can act as a natural filter capturing pollutants (hydrocarbons and heavy metals) and microorganisms washed-off by runoffs (e. g., Murphy and Ginn, 2000; Tedoldi et al., 2016). Nevertheless, the effectiveness of SIS in preventing the migration of contaminants towards aquifers is not always optimal (Borchardt *et al.*, 2007; Lapworth et al., 2012; Arnaud et al., 2015; Voisin et al., 2018). The filtering properties of SIS are influenced by various abiotic factors such as the nature of the media (rocks, sand and other soil elements), the physical properties (e. g., granulometry, hydrophobicity index, organization), and the runoff water flow velocity (Lassabatere et al., 2006; Winiarski et al., 2013). These constraints will impact both the water transit time from the top layers to the aquifer, but also the biological properties of these systems including the plant cover, root systems, worm population and composition of microbiota (Barba et al., 2019; Bedell et al., 2013; Crites, 1985; Pigneret et al., 2016). The thickness of the vadose zone was found to be one of the key parameters explaining chemical transfers such as phosphate and organic-carbon sources (Voisin et al., 2018). The situation is much less clear regarding the microbiological communities that flow through these systems (e. g., Barba et al., 2019; Voisin et al., 2018).

According to the microbial community coalescence concept conceptualized by Tikhonov, (2016) and adapted to riverine networks by Mansour et al. (2018), urban aquifers fed by SIS should harbor microbiota reflecting the coalescence (community assemblages and selective sorting) of aboveground microbial communities with those of the aquifer. Indeed, during rain events, microbial communities will be re-suspended through runoff-driven surface erosion processes, favoring detachment of microorganisms from plant litter, wastes, soil, and other particles (Mansour et al. 2018). These re-suspended communities will merge and generate novel assemblages. The resulting community will initially match the relative contributions of the various sub-watersheds to the overall microbiological complexity of the assemblages (Mansour et al. 2018). The prevailing ecological constraints among the downward systems will then gradually drive this coalescence towards the most fit community structures. These resulting communities might be highly efficient at degrading urban pollutants trapped among a SIS, but could also disturb the ecological equilibria of the connected and more sensitive systems like those of deep aquifers as suggested by Voisin et al. (2018).

This study explores the impact of a SIS with a thick vadose zone (> 10 m) on the coalescence of urban runoff microbial communities in a connected aquifer. Two hypotheses were tested (1) highly specialized taxa (often

termed K-strategists e. g., Vadstein et al., 2018) of an aquifer outcompete the intrusive community members of
aboveground taxa; and (2) nutrient inputs from runoffs and pollutants drive changes among communities and
favour environmental opportunists (often termed r-strategists e. g., Vadstein et al., 2018). The targeted SIS is part
of a long-term experimental site (http://www.graie.org/portail/dispositifsderecherche/othu/) that records both
physico-chemical and biological properties. This SIS is connected to the eastern aquifer of Lyon (France) which
is fed by three low hydraulic conductivity corridors ($10^{-5}$–$10^{-8}$ m. s$^{-1}$) separated by moraine hills (Foulquier et al.,
2010). The average vadose zone thickness of the SIS is 15 m, and the delay between a rainfall event and the impact
on the aquifer waters was estimated at 86±11h (Voisin et al., 2018). A 16S rRNA gene meta-barcoding dataset
was assembled for this site, to investigate bacterial community coalescence from the top compartments into the
connected aquifer waters but also the biofilm communities developing on inert surfaces. This investigation was
also built on the hypothesis that a less significant microbial community coalescence is expected in aquifer waters
than biofilms. This is supported by previous reports which suggested the occurrence of transient free-living
bacteria among aquifers acting as a traveling seed bank (Griebler et al., 2014). For such monitorings, water grab
samples were previously found to give access to snapshots of the diversity found within an aquifer (Voisin et al.,
2018) whereas aquifer biofilms developing on artificial surfaces (clay beads) were shown to be more integrative
and informative of the groundwater microbiological quality (Mermillod-Blondin et al., 2019). Clay bead biofilms
were found to capture the most abundant aquifer taxa, and taxa that could not be detected from grab samples.
However, some bacterial taxa were still not detectable by this approach because of a poor ability at colonizing clay
beads over short time periods. A field based investigation was thus performed to further explore the relative
contributions of a set of sources such as runoffs and urban soils on the observed biofilm assemblages recovered
from an aquifer. A Bayesian methodology, named SourceTracker (Knights et al., 2011), was used to investigate
community coalescence from 16S rRNA gene – based DNA meta-barcoding datasets. Complementary datasets
were then assembled from an additional DNA marker named *tpm* (encoding EC:2.1.1.67 which catalyzes the
methylation of thiopurine drugs) (Favre-Bonté et al., 2005). This genetic marker enables finer taxonomic
allocations down to the species level to explore the coalescence of a set of waterborne bacterial species and sub-
species, including plant and human pathogens, within the aquifer microbial community.

## 2  Material and Methods

### 2.1  Experimental site

The Chassieu urban catchment is located in the suburbs of Lyon (France). It has a surface of 185 ha and hosts
mainly industrial and commercial activities (*i.e.* wholesaling, recovery and waste management, metal surface
treatment, car wash and repair services). The imperviousness coefficient of the catchment area is approximately
75%. No significant modifications impacting the urban watershed were recorded during the investigation.
Stormwater and dry weather flows from industrial activities are drained by a network separated from the sewer.
This network transfers waters into the Django-R SIS, which is a part of the OTHU long term experimental
observatory dedicated to urban waters (http://www.graie.org/othu/). This SIS contains an open and dry detention
basin (DB) (32,000 m$^3$) built on a concrete slab, with edges impermeabilized by a thick plastic lining. This DB
allows a settling of coarse and medium size particles, resulting in sedimentary deposits which favor plant cover
development. The DB water content is delivered within 24h into an infiltration basin (IB) (61,000 m$^3$), which
favors the recharge of the connected aquifer (AQ). This infiltration basin had a vadose zone of about 11 m during
the investigation, and its geology, hydrology, ecology and pollution levels had been previously investigated (e. g.,
Barraud et al. 2002; Le Coustumer and Barraud, 2007; El-Mufleh et al. 2014).
The Chassieu watershed, the Django-R SIS, and the Lyon aquifer were investigated in this study (Figure 1,
Table S1). Watershed runoff waters (hereafter WS) have been collected from sampling points spread over the
catchment (21 sub-watersheds over three sampling periods, n=64 samples). Sediments from the detention basin
(hereafter DB) were recovered from a 50 cm$^2$ area covering the full sediment column down to the concrete slab
(n=20 samples). These sediments (or urban soils) often had an herbaceous plant cover and were sampled in four
areas defined according to the hydrological forces prevailing in the basin (e. g., Marti et al., 2017; Sébastian et al.,
2014). Infiltration basin soil samples (hereafter IB) were collected from 3 zones (the area receiving the inflow
waters, the bottom area of the basin, and an upper zone of the basin exposed to inflow waters only during heavy
rain events) (n=5 samples per zone) at a 0-10 cm depth covering a surface of 50 cm$^2$. The aquifer samples were
recovered from piezometers located upstream (up, in a zone of the aquifer not influenced by water recharge) and
downstream (dw, in a zone of the aquifer influenced by water recharge) of the SIS of the Django-R site at a depth
of 2 m below the water table (e. g., Barraud et al., 2002; Voisin et al., 2018) (Figure 1). Groundwater samplings
(n=6; named AQ_wat) were performed with an immerged pump, used at a pumping rate of 6–8 L. min$^{-1}$ (PP36
inox, SDEC, Reignac-sur-Indre, France), and previously cleaned with 70 % ethanol. The first 50 L were used to
rinse the sampling equipment and were subsequently discarded. The following 6 L were used for the
microbiological analyses. The biofilm samples (AQ_bio) from the aquifer were recovered from clay beads
incubated in the aquifer over 10 days using the piezometers described above (n=6 samples). Clay beads were used
as physical matrices to sample groundwater biofilms according to Voisin et al. (2016).

**2.2  DNA extractions, 16S rRNA gene qPCR assays, and PCR products DNA sequencings**

Approximately 600 mg of sediments or soils, or up to 5 L of aquifer or runoff water samples filtered using 0.22
μm polycarbonate filters, were used per DNA extraction. Total DNA was extracted from soils/sediments or filters
using the FastDNA SPIN® Kit for Soil (MP Biomedicals, Carlsbad, France). For clay bead biofilms, microbial
cells were detached by shaking at 2500 rpm for 2 min in 10 mL of 0.8 % NaCl. These suspensions were then
filtered and their DNA contents were extracted as indicated above. Blank samples were performed during these
extractions for both the soils/sediments or filtered cells. DNA was quantified using a nanodrop UV-Vis
spectrophotometer. Blank DNA extracts showed values below the detection limit. DNA extracts were visualized
after electrophoresis at 6V. cm$^{-1}$ using a TBE buffer (89 mM Tri-borate, 89 mM boric acid, 2 mM EDTA, (pH
8.0)) through a 0.8 % (w/v) agarose gel, and DNA staining with 0.4 mg. mL$^{-1}$ ethidium bromide. A Gel Doc XR+
System (Bio-Rad, France) was used to observe the stained DNA, and confirm their relative quantities (between
20-120 ng. μL$^{-1}$; median value around 40 ng. μL$^{-1}$) and qualities. DNA was kept at -80°C and shipped on ice within
24h to the DNA sequencing services when appropriate.
Quantitative PCR assays were performed on the DNA extracts to estimate their relative content in 16S rRNA gene
copies. These assays were performed on a Bio-Rad CFX96 realtime PCR instrument with Bio-Rad CFX Manager
software, version 3.0 (Marnes-la-Coquette, France). The 16S rRNA gene primers 338F and 518R described by
Park and Crowley (2006) were used, together with the Brilliant II SYBR green low ROX qPCR master mix for
SYBR Green qPCR. Melting T° was 60°C. Linearized plasmid DNA containing a 16S rRNA gene was used as a
standard, and obtained from Marti et al. (2017). Presence of inhibitors in the DNA extracts was checked by spiking
known amount of plasmids harboring *int2* ($10^7$ copies of plasmid per μL) in the PCR mix. Number of cycles needed
to get a PCR signal was compared with wells where only plasmid DNA harboring *int2* was added to the qPCR
mix. When a high number of cycles was needed to observe a signal, a 5- or 10-fold dilution of the DNA extract
was done, and another round of tests was performed to confirm the absence of PCR inhibitions. Each assay was
triplicated on distinct DNA extracts, and technical triplicates were performed. The 16S rRNA gene qPCR datasets
are presented in Figure S1. These assays confirmed the high number of bacterial cells per compartment (Figure S1
and Table S2): (1) soils from the infiltration basin (IB) had a median content of $1.32 \times 10^{11}$ 16S rRNA gene copies
per g dry weight; (2) sediments from the detention basin (DB) of $1.83 \times 10^{11}$ 16S rRNA gene copies per g dry
weight, (3) the runoff waters (WS) had a median content of $4.75 \times 10^8$ 16S rRNA gene copies per mL, (4) the
aquifer waters (AQ_wat) of $3.10 \times 10^6$ 16S rRNA gene copies per mL, and (5) the aquifer clay bead biofilms
showed $1.35 \times 10^7$ 16S rRNA gene copies per cm$^2$.
Sequencing of V5-V6 16S rRNA gene (*rrs*) PCR products were performed by MrDNA DNA sequencing services
(Shallowater, Texas, USA) on an Illumina Miseq V3. The PCR products were generated using DNA primers 799F
(barcode + ACCMGGATTAGATACCCKG) and 1193R (CRTCCMCACCTTCCTC) reported by Beckers et al.
(2016). PCR amplifications were performed using the HotStarTaq Plus Master Mix Kit (Qiagen, USA) using the
following temperature cycles: 94 °C for 3 min, followed by 28 cycles of 94 °C for 30 s, 53 °C for 40 s, and 72 °C
for 1 min, with a final elongation step at 72 °C for 5 min. PCR products and blank control samples were verified
using a 2 % agarose gel and following the electrophoretic procedure described above. PCR products obtained from
field samples showed sizes around 430 bp but blanks did not show detectable and quantifiable PCR products.
Dual-index adapters were ligated to the PCR fragments using the TruSeq® DNA Library Prep Kit which also
involved quality controls of the ligation step (Illumina, Paris, France). Illumina Miseq DNA sequencings of the
PCR products were paired-end, and set up to obtain around 40K reads per sample.
The *tpm* DNA libraries were also sequenced by the Illumina MiSeq V3 technology but by the Biofidal DNA
sequencing services (Vaulx-en-Velin, France). PCR products were generated using the following mix of
degenerated PCR primers: ILMN-PTCF2 (5'- P5 adapter tag + universal primer +
GTGCCGYTRTGYGGCAAGA-'3), ILMN-PTCF2m (5'- P5 adapter tag + universal primer +
GTGCCCYTRTGYGGCAAGT-'3), ILMN-PTCR2 (5'- P7 adapter tag + universal primer +
ATCAKYGCGGCGCGGTCRTA-'3), and ILMN-PTCR2m (5'- P7 adapter tag + universal primer +
ATGAGBGCTGCCCTGTCRTA-'3) targeting conserved regions defined by Favre-Bonté et al. (2005). The
universal primer was 5'-AGATGTGTATAAGAGACAG-'3. The P5 adapter tag was : 5'-TCGTCGGCAGCGTC-
'3. The P7 adapter tag was : 5'- GTCTCGTGGGCTCGG-'3. PCR reactions were performed using the 5X Hot
BIOAmp® master mix (Biofidal, France) containing 12.5 mM $MgCl_2$, and 10 % DMSO and 50 ng sample DNA
final concentrations. PCR cycles were as follow: (1) a hot start at 94°C for 5 min, (2) 35 cycles consisting of 94°C
for 30 s, 58°C for 30 s and 72°C for 30 s, and (3) a final extension of 5 min at 72°C. The mix had two carefully
optimized enzymes, the HOT FIREPol® DNA polymerase and a proofreading polymerase. This enzyme blend
has both 5'→ 3' exonuclease and 3'→ 5' proofreading activities. This mix exhibits an increased fidelity (up to
five fold) compared to a regular *Taq* polymerase. PCR products and blank control samples were verified using a
2% agarose gel and following the electrophoretic procedure described above. PCR products obtained from field
samples showed sizes around 320 bp but blanks did not show detectable and quantifiable PCR products. Index and
Illumina P5 or P7 DNA sequences were added by Biofidal through a PCR procedure using the same Hot

BIOAmp® master mix and the above temperatures, but limited to 15 PCR cycles. Indexed P5/P7 tagged PCR products were purified using the SPRIselect procedure (Beckman Coulter, Roissy, France). PCR products and blank control samples were verified using the QIAxcel DNA kit (Qiagen, France), and band sizes around 400 bp were observed but not in the blank samples. Quantification of PCR products by the picogreen approach using the Quantifluor dsDNA kit (Promega, France) and a Qubit® 2.0 Fluorometer (Thermo Fisher Scientific, France) was performed, and showed low values among the blanks which were at the limit of detection (around 0.07 ng. $\mu L^{-1}$). Still, *tpm* harboring bacteria being in low number among a bacterial community (about 2-3%), these controls were run during the Miseq DNA sequencing of the PCR products. Illumina Miseq DNA sequencings of the *tpm* PCR products were paired-end, and set up to obtain around 40K reads per sample. Blank samples generated low numbers of *tpm* reads (blank 1 = 24 reads; blank 2 = 3 reads, blank 4 = 1028 reads, and blank 5 = 1 read), and these have been listed in Table S3. These reads mainly belonged to unknown species (86%). However, reads from *P. fluorescens* (from OTUs not found in the field samples), *P. xanthomarina* (17 reads over all blanks) and *P. fragi* (n=3 reads over all blanks) were recovered but did not have any impact on the coalescence analysis. The 16S rRNA and *tpm* gene sequences reported in this work are available at the European Nucleotide Archive (https://www.ebi.ac.uk/ena).

### 2.3 Bioinformatic analyses

All paired-end MiSeq reads were processed using Mothur 1.40.4 by following a standard operation protocol (SOP) for MiSeq-based microbial community analysis (Schloss et al., 2009; Kozich *et al.*(2013), so-called MiSeq SOP and available at http://www.mothur.org/wiki/MiSeq_SOP. Due to the large number of sequences to be processed, the cluster.split command was used to assign sequences to OTUs. For the 16S rRNA (*rrs*) gene sequences, reads were filtered for length (>300bp), quality score (mean, ≥25), number of ambiguous bases (=0), and length of homopolymer runs (<8) using the trim.seqs script in Mothur, and singletons were discarded. The 16S rRNA gene sequences passing these quality criteria were aligned to the SILVA reference alignment template (release 128). Unaligned sequences were removed. Chimeric sequences were identified using the chimera.uchime command and removed. Variability in the number of cleaned reads per sample was observed but not correlated with variations in the number of 16S rRNA gene sequences (Table S2). These variations were thus considered to be due to the DNA sequencing process. Therefore, a sub-sampled dataset (20,624 reads per sample; with exclusion of samples with total reads below this threshold) was used to mitigate the artifact of sample library sizes. Operational Taxonomic Units (OTUs) were defined using a 97% identity cut-off as recommended by several authors in order to collapse sequences into groups that reduce the incidence of sequence errors on the datasets (e. g., Eren et al. 2013; and Johnson et al. 2019). It is to be noted that amplicon sequence variants (ASV) could also be used to build contingency tables (e. g., Callahan et al. 2016; Karstens et al. 2019). However, exact sequence variants can generate uncertainties when using 16S rRNA gene sequences because of variations among species and strains due to the presence of multiple copies per genome (Johnson et al. 2019). Figure S2 shows the OTU rarefaction curves for the full and the sub-sampled datasets. The sub-sampled dataset was used for all downstream analyses except those of the SourceTracker Bayesian approach (see below). OTUs were affiliated to taxonomic groups by comparison with the SILVA reference alignment template if a bootstrap P-value over 80% was obtained. FAPROTAX (Louca et al., 2016) functional inferences were performed on the MACADAM Explore web site (http://macadam.toulouse.inra.fr/) according to Le Boulch et al. (2019). For the *tpm* gene sequences, chimeric sequences, primers, barcodes were removed, and the dataset was limited to sequences of a minimum length of 210

bp (average length=215 bp). These reads were aligned against the *tpm* database (BD_TPM_Mar18_v1.unique_770seq). Unaligned sequences were removed. The number of sequences per sample was then sub-sampled (4,636 sequences per sample; with exclusion of samples with total reads below this threshold). Operational Taxonomic Units (OTUs) were defined at a 100% identity cut-off. The "BD_TPM_Mar18_v1.unique_770seq" database (http://www.graie.org/othu/donnees) was used to classify the sequences using the "Wang" text-based Bayesian classifier (Wang et al., 2007) and a P-bootstrap value above 80%. Local BLAST analyses were performed on the "BD_TPM_Mar18_v1.unique_770seq" database using the NCBI BLASTX program to check the quality of the taxonomic affiliations.

**2.4  Statistical analyses**

All statistical analyses were performed in R (v.3.5.1). For the 16S rRNA gene sequences, alpha-diversity estimates were computed using the function "rarefy" from the 'Vegan' package (Oksanen et al., 2015). Richness ($S_{obs}$) was computed as the number of observed OTUs in each sample. The diversity within each individual sample was estimated using the non-parametric Shannon index. To estimate whether the origin of the samples influenced the alpha-diversity, an ANOVA with Tukey's post-hoc tests was performed. Shared and unique OTUs were depicted in Venn-diagrams with the "limma" package (Ritchie et al., 2015). Concerning the beta-diversity analyses, a neighbor-joining tree was constructed with a maximum-likelihood approximation method using FastTree (Price et al., 2009). Weighted UniFrac distances were calculated for all pairwised OTU patterns according to Lozupone et al. (2011), and used in a Principal Coordinates Analysis (PCoA) (Anderson and Willis, 2003). Permutation tests of distances (PERMANOVA) (Anderson, 2001) were performed using the "vegan" package (Oksanen et al., 2015), to establish the significance of the observed groupings.

**2.5  Bacterial community coalescence analyses**

The SourceTracker computer package (Knights et al., 2011) was used to investigate community coalescence. SourceTracker is a Bayesian approach built to estimate the most probable proportion of user-defined "sources" DNA reads in a given "sink" community. In the present analysis, various scenarios of community coalescence were investigated such as the coalescence of bacterial taxa from the watershed runoff waters and sediments from the detention and infiltration basins with those of the downstream SIS aquifer water samples or of recent biofilms developing on clay beads incubated in the aquifer. SourceTracker was run with the default parameters (rarefaction depth = 1000 reads from the original cleaned dataset of 16S rRNA gene reads (Fig. S2a), burn-in: 100, restart: 10) to identify sources explaining the OTU patterns observed among the aquifer samples (waters and clay bead biofilms, n=12). Alpha values were tuned using cross-validation (alpha 1= 0.001 and alpha 2= 1). The relative standard deviation (RSD) based on three runs was used as a gauge to evaluate confidence on the computed values (Henry et al., 2016; McCarthy et al., 2017).

**3  Results**

**3.1  16S rRNA V5-V6 gene sequences distribution biases and profilings**

The analysis of the 16S rRNA V5-V6 gene libraries yielded 2,124,272 high-quality sequences distributed across 103 samples as described in Table S2. Subsampling-based normalization was applied (20,624 reads per sample) and sequences were distributed into 10,231 16S rRNA gene OTUs (with > 97 % identity between sequences of an OTU). The rarefaction curves are shown in Figure S2. At all sampling sites, bacterial communities were dominated

by Proteobacteria, Bacteroidetes and Actinobacteria (WS=95 % of total reads, DB=84 %; IB=71 %; AQ_bio=99 % and AQ_wat=59 %), but 10 other phyla with relative abundances greater than 0.5 % were also detected (Figure 2A and Table S4). Alpha-diversity estimates showed that aquifer samples harbored a microbiome with a significantly lower richness (AQ_bio: $S_{obs}$=278 OTUs ± 106 and AQ_wat: $S_{obs}$=490 OTUs ± 333) and a less diverse bacterial community (AQ_bio: H'=2.9 ± 0.3 and AQ_wat: H'=4.3 ± 0.7) than the ones of the upper compartments ($S_{obs-WS}$=1,288 OTUs ± 232; $S_{obs-DB}$=1,566 OTUs ± 245, $S_{obs-IB}$=1,503 OTUs ± 177 and H'$_{WS}$=5.0 ± 0.5; H'$_{DB}$=5.4 ± 0.5, H'$_{IB}$=5.7 ± 0.4) (ANOVA, p<0.001) (Figure 2B and Table S5). Among the surface samples, a greater diversity was observed among the soil samples from the infiltration basin than from samples of watershed runoff waters and sediments recovered from the detention basin (ANOVA, p<0.05). In the aquifer, water grab samples were more diverse and showed higher 16S rRNA gene OTU contents than biofilms recovered from the clay beads incubated over a 10-day period (ANOVA, p<0.05) (Figure 2B and Table S5).

The structure of bacterial communities inferred from V5-V6 16S rRNA gene sequences changed markedly along the watershed. A PCoA ordination of the OTU profiles based on weighted Unifrac distances showed samples to be clustered according to their compartment of origin (*i.e.* WS, DB, IB, AQ_bio and AQ_wat) (Figure 3). These changes in community structures between compartments were supported by PERMANOVA statistical tests (F=20.7, P<0.001). Bacterial communities per compartment were found to contain core and flexible (defined as not conserved between all sampling periods) bacterial taxa. Within the same compartment, similarities between bacterial community profiles ranged from 65 % (AQ_wat) to 82 % (IB), whereas similarities across compartments ranged from 48 % (DB vs AQ_bio) to 66 % (DB vs IB) (Figure S3). Bacterial community profiles of the aquifer waters were found closer to the ones of the detention basin deposits (57 %) and soils of the infiltration basin (61 %) than those of the aquifer biofilms (48 and 49 %, respectively). However, more than 89 % of the 16S rRNA gene OTUs (n=8,284) identified above the aquifer (WS, DB and IB) were not detected in groundwater samples (AQ_bio and AQ_wat) (Figure S4). This large group of OTUs was made of minor taxa which accounted for 37 %, 44% and 47% of the total reads recovered from the WS, DB and IB samples, respectively.

**3.2  Coalescence of surface and aquifer bacterial communities**

A SourceTracker analysis was performed to estimate the coalescence of bacterial taxa inferred from V5-V6 16S rRNA gene reads from the watershed and SIS down into the aquifer waters and biofilm bacterial communities. This analysis indicated significant coalescence between the bacterial communities of the runoffs, the soils/sediments of the SIS, and the aquifer samples. The aquifer water microbial community upstream the SIS was found to explain about 40 % of the downstream water microbial community (Table 1), while 16S rRNA gene reads from the runoff waters were found to explain about 5 %, and those of the DB around 8 % of the observed patterns (Table 1). The infiltration basin explained about 7 % of the observed diversity among the SIS impacted aquifer water community. The aquifer biofilm bacterial communities were also found to be assemblages of communities from the surface environments. The origin of more than 94 % of the SIS impacted aquifer biofilms could be explained by the SourceTracker. Main sources of taxa were inferred to be the upstream aquifer waters (59 %), the sediments of the detention basin (22 %), and the runoff waters (8.5 %) (Table 1). Soils from the infiltration basin did not appear to have contributed substantially to the taxa recovered from these aquifer biofilms (<4 %) (Table 1). Aquifer biofilms recovered upstream the SIS showed a high proportion of taxa related to those observed among the runoff waters (44 %) and the aquifer waters (49 %). This was not considered surprising because runoff

320 infiltration can occur in several sites upstream of the SIS (although no direct relation with other SIS could be made
321 so far).

### 3.3. 16S rRNA gene inferred bacterial taxa undergoing coalescence in the aquifer

323 To identify the bacterial taxa involved in the coalescence process, OTUs of the 16S rRNA gene dataset were
324 allocated to taxonomic groups using the SILVA reference alignment template. These taxonomic allocations
325 indicated that (1) 14 genera were only recorded in the aquifer samples, (2) 421 genera were only recorded in the
326 upper surface compartments of the watershed, and (3) 219 were recorded among aboveground and aquifer
327 compartments (Table S6). The following bacterial genera were exclusively associated to the aquifer bacterial
328 communities: *Turicella, Fritschea, Metachlamydia, Macrococcus, Anaerococcus, Finegoldia, Abiotrophia,*
329 *Dialister, Leptospirillum, Omnitrophus, Campylobacter, Sulfurimonas, Haemophilus,* and *Nitratireductor*. These
330 bacterial genera were recovered from all water samples, and 5 were also detected in biofilms (Table S6). These
331 genera were associated to 926 16S rRNA gene OTUs that accounted for, respectively, 48 % and 1.8 % of the total
332 reads recovered from aquifer waters and aquifer biofilms developing on clay beads. FAPROTAX functional
333 inferences indicated some of these genera to be host-associated such as *Fritschea, Metachlamydia, Finegoldia,*
334 *Campylobacter* and *Haemophilus*, with the latter two being well-known to contain potential pathogens.
335 *Campylobacter* and *Sulfurimonas* cells have also been associated with nitrogen and sulfur respiration processes,
336 and *Leptospirillum* with nitrification.

337  Regarding the bacterial taxa of the aboveground communities matching those of the aquifer samples, a total of
338 1,021 16S rRNA gene OTUs was found to be shared between these compartments (Table 2 and Figure S4). These
339 OTUs consisted of abundant taxa as they accounted for 9.7-39.4 % of the total reads for the samples recovered
340 from the surface compartments, and for 33.6-83.4 % and 95.0-99.4 % of the total reads of the water and biofilm
341 aquifer samples, respectively (Table 2). The β- and γ-proteobacteria dominated this group. It is noteworthy that
342 aquifer samples collected upstream of the SIS shared less OTUs with the surface compartments (125 OTUs ± 41)
343 than samples under the influence of the infiltration system (332 OTUs ± 85) (Table 2 and Figure S4). The shared
344 OTUs between aquifer samples and the upper compartments represented a higher fraction of bacterial communities
345 in samples recovered downstream of the SIS (81.3 % ± 22.8 of total reads) compared to those collected upstream
346 (68.9 % ± 30.9 of total reads) (Table 2). Reads from *Pseudomonas, Nitrospira, Neisseria, Streptococcus*, and
347 *Flavobacterium* were the most abundant (>1 %) of the shared OTUs recovered in the aquifer water samples,
348 whereas those allocated to *Pseudomonas, Duganella, Massilia, Nocardia, Flavobacterium, Aquabacterium,*
349 *Novosphingobium, Sphingobium, Perlucidibaca,* and *Meganema* were the most abundant (>1 %) among the
350 aquifer biofilms (Table S6). Most of these aquifer water taxa (except *Streptococcus*) were found to be involved in
351 denitrification or nitrification as inferred from FAPROTAX. The biofilm taxa were most often associated with
352 hydrocarbon degradation (*Novosphingobium, Sphingobium,* and *Nocardia)* by FAPROTAX. Several of these
353 biofilm bacterial genera were also found to be containing potential human pathogens (*Duganella, Massilia,*
354 *Nocardia*, and *Aquabacterium*) by FAPROTAX (and published clinical records). A set of 14 potentially hazardous
355 bacterial genera was selected from Table S6, and used to illustrate the coalescence of bacterial taxa among the
356 aquifer samples on Figure 4. The 16S rRNA gene reads from *Flavobacterium* prevailed in all upper compartments
357 (WS=6.9 % of total reads, DB=13.4 % and IB=8.3 %) and were in significant numbers among the connected
358 aquifer (AQ_wat = 1.1 % and AQ_bio = 3.1 %) (Figure 4B and Table S6). *Pseudomonas* 16S rRNA gene reads

were in relatively lower numbers in the upper compartments (WS = 0.4 % of total reads, DB = 0.4 % and IB <
0.05 %) than the aquifer (AQ_wat = 8.4 % and AQ_bio = 35.5 %) (Figure 4B and Table S6). Similar trends were
observed for *Nocardia* and *Neisseria* OTUs (Figure 4B). Notably, OTUs exclusively recovered from the upper
compartments were mainly allocated to the *Gemmatimonas* (0.2-1.6 % of total reads), *Geodermatophilus* (0.1-1.8
%) and *Roseomonas* (0.1-1.0 %) (Table S6).
**3.4 Coalescence of *Pseudomonas* and other *tpm*-harboring bacterial species**
DNA sequences from *tpm* PCR products generated according to Favre-Bonté et al. (2005) allowed a further
exploration of the bacterial species undergoing a coalescence with the aquifer microbiome. A total of 19,129 *tpm*
OTUs was recorded among the samples (from datasets re-sampled to reach 4,636 reads per sample). As expected,
these *tpm* reads were mainly assigned to the *Proteobacteria* (WS = 92 % of total reads, DB = 87 %; IB = 76 %;
AQ_wat = 83 % and AQ_wat = 85 %), but some reads could also be attributed to the *Bacteroidetes*, *Nitrospirae*
and *Cyanobacteria* (Table S7). These taxonomic allocations allowed the identification of 24 bacterial genera and
91 species whose distributions are summarized in Tables S7 and S8. The *tpm* sequences were mainly allocated to
*Pseudomonas* (WS = 36 % of the reads, DB = 27 %; IB = 7 %; AQ_wat = 51 % and AQ_bio = 48 %), *Aeromonas*
(WS = 1 % of the reads, DB = 3 %; IB <0.05 %; AQ_wat = 0.07 % and AQ_bio < 0.05 %), *Xanthomonas* (WS =
4 % of the reads, DB <0.05 %; IB =1 %; AQ_wat = 8 % and AQ_bio < 0.05 %), *Herbaspirillum* (WS = 11 % of
the reads) and *Nitrosomonas* (DB = 4 % of the reads; IB = 0.2 %) (Table S8). Reads related to *Pseudomonas* were
allocated to 50 species, including pollutant-degraders (*P. pseudoalcaligenes, P. aeruginosa, P. fragi, P.*
*alcaligenes, P. putida* and *P. fluorescens*), phytopathogens (*P. syringae, P. viridiflava, P. stutzeri,* and *P.*
*marginalis*) and human opportunistic pathogens (*P. aeruginosa, P. putida, P. stutzeri, P. mendocina, S.*
*acidaminiphila*) (Table S9). It is to be noted that blank samples sequenced during the *tpm* meta-barcoding assay
revealed 23 *Pseudomonas* OTUs coming from the DNA extraction kit or generated during the PCR product
Illumina sequencing process (Table S3). Only OTU00573 was found in high number (867 reads), but this
contaminant did not have an impact on the coalescence analysis because of its absence in the below ground datasets
(Table S10). Other contaminant OTUs did not represent more than 10 times the ones observed in the field samples
for identical OTUs, a criterion used to distinguish significant contaminants (Lukasik et al., 2017). In fact, only
seven OTUs found among the blanks matched OTUs recovered from the environmental samples, and only two of
these could be related to well-defined species *i. e., P. xanthomarina* (17 reads among all blanks) and *P. fragi* (three
reads among all blanks). These reads matched a single OTU over eleven allocated to *P. xanthomarina* in the
environmental samples, and one OTU over 52 for *P. fragi* (Table S10). Reads related to the *Aeromonas* were
attributed to 11 species, but only those allocated to *A. caviae* could be recovered from the aquifer and aboveground
compartments (Table S9). Reads related to the *Xanthomonas* were allocated to 9 species, but only those allocated
to the *X. axonopodis/campestris* complex and *X. cannabis* species were recovered from the aquifer and upper
compartments (Table S9). Regarding the *Pseudomonas*, *tpm* reads allocated to *P. jessenii, P. chlororaphis,* and *P.*
*resinovorans* were restricted to the aquifer samples. Reads allocated to *P. aeruginosa, P. anguilliseptica, P.*
*chengduensis, P. extremaustralis, P. fluorescens, P. fragi, P. gessardii, P. koreensis, P. pseudoalcaligenes, P.*
*putida, P. stutzeri, P. umsongensis,* and *P. viridiflava*, were recovered from the aquifer and upper compartments
(Table S9). FAPROTAX analysis indicated that a significant number of the species detected in the aquifer can be
involved in denitrification (*P. aeruginosa, P. fluorescens, P. putida, P. stutzeri, S. acidaminiphila, X.*
*autotrophicus, P. chlororaphis*) or nitrification (*Nitrospira defluvii, Nitrosomonas oligotropha*) but also in
hydrocarbon degradation (*P. aeruginosa, P. fluorescens, P. putida*). Some of these species were also suggested by
FAPROTAX to be human pathogens or invertebrate parasites (e. g., *P. chlororaphis, P. aeruginosa*).
The *tpm* OTUs (representative of infra-specific complexes) shared between the upper compartments and the
aquifer were allocated to 14 species and 5 genera (Table 3 and Table S10). Four of these OTUs led to higher
relative numbers of reads in the aquifer samples, in the following decreasing order: *P. umsongensis* (Otu00005) >
*P. chengduensis* (Otu00024) > *X. axonopodis/campestris* (Otu00019 & Otu00878) > *P. stutzeri* (Otu00119 &
Otu10066). These co-occurrences of OTUs between aboveground and aquifer samples support the hypothesis of
significant coalescence between these bacterial communities. The other OTUs showed higher number of reads
among the top compartments. The OTU allocated to *X. cannabis* showed the highest relative number of reads of
this group among runoff waters. The distribution pattern of this OTU suggested a relative decline when moving
down the aquifer. The *P. aeruginosa* Otu00066 was recovered in the runoff waters, and biofilms developing on
clay beads incubated in the aquifer.

## 4  Discussion

The coalescence of bacterial taxa from runoff and stormwater infiltration systems (SIS) with those of a connected
aquifer was investigated. Taxonomic and functional inferences were performed using 16S rRNA gene libraries. In
addition, a genetic marker named *tpm* was used to track species and particular sequence types of the *Pseudomonas*,
*Aeromonas*, and *Xanthomonas* (and a few other genera) from runoffs down into the SIS impacted aquifer.
Estimation of alpha-diversity indices from the 16S rRNA bacterial community profilings indicated that
groundwater samples (*i.e.* waters and biofilms) harbored a less diverse microbiome than those of the top
compartments (*i.e.* WS, DB, IB). A 2 to 5-fold reduction in bacterial richness was observed from the surface
compartments down into the aquifer. This result suggested that a high proportion of bacterial taxa carried by
stormwater runoffs or thriving in the detention/infiltration basins were retained and/or eliminated by the vadose
zone filtration process. In line with this result, the estimation of the copy number of the bacterial 16S rRNA gene
by qPCR revealed that bacterial biomass was much lower in aquifer than in runoff samples. In fact, more than 89
% of the 16S rRNA gene OTUs in the top compartments were not detected in the underground samples. This is in
agreement with previous works which have shown that immobilization of microorganisms through porous media
are high in the top soil layers and triggered by mechanical straining, sedimentation and adsorption (Kristian Stevik
et al., 2004; Krone et al., 1958). Moreover, particles that accumulate as water passes through the soil can form a
mat that enhances this straining process (Krone et al., 1958). Despite this filtering effect, infiltration induces
significant changes in the diversity of groundwater bacterial communities. Both, water and biofilm aquifer samples
recovered downstream the SIS had higher bacterial richness than those collected upstream. It is to be noted that
soils of the infiltration basin showed higher bacterial diversity than those of the sediments of the detention basin
and runoffs. This is most likely related to a development of plant-associated bacteria in this compartment. Indeed,
the infiltration basin was covered by several plant species of *Magnoliophyta* like *Rumex* sp. which can disseminate
rapidly through rhizomes (Bedell et al., 2013) and generate multiple ecological niches for bacteria.
The SourceTracker Bayesian probabilistic approach based on 16S rRNA gene meta-barcoding datasets
(Knights et al., 2011) was applied to refine our understanding of the coalescence of microbial communities from
aboveground environments down into an aquifer. These inferences revealed variable levels of coalescence in the
SIS recharged aquifer depending upon the investigated sink *i.e.* waters or biofilms developing on clay beads

incubated in the aquifer. Bacterial community structures of the groundwater samples (upstream and downstream the SIS) were significantly built from aboveground communities (*e. g.,* those from runoff waters). However, the origin of a high proportion of the diversity observed among the aquifer waters downstream the SIS remained undefined. This is likely related to the emergence of novel biomes among the vadose zone of a SIS fed with urban waters and pollutants. These biomes would have emerged from the build-up of novel biotopes during the construction and functioning of the SIS (see Winiarski (2014) for review). The prevailing environmental constraints and pollutants would then have favored minor taxa (not detectable by meta-DNA barcoding approaches) from the aboveground compartments. In fact, chemical pollutants have been shown to be significantly washed-off or transported with particles during rain events (El-Mufleh et al., 2014), and some of these were found to reach aquifers fed by SISs (Pinasseau et al. 2019). Among these pollutants, Bernardin-Souibgui et al. (2018) reported that urban sediments found in the detention basin of the experimental site were heavily polluted by polycyclic aromatic hydrocarbons (PAH). Their contents were estimated at $197\pm36$ ng. g dw$^{-1}$ (dry weight) for light PAHs, and at $955\pm192$ ng. g dw$^{-1}$ for heavy PAHs. PCBs were also recorded for the seven congeners of the European norm for a total of 0.2 to 2.1 mg. kg dw$^{-1}$ (Sebastian et al., 2014). Metallic trace elements (MTE) were recorded in significant amounts, with Cu being found at about 280 mg. kg dw$^{-1}$, Pb at about 200 mg. kg dw$^{-1}$, Zn at about 1600 mg, and Cd at about 5 mg. kg dw$^{-1}$ (Sebastian et al., 2014). MTE, PCBs and PAHs were also recorded in the soils of the infiltration basin at similar concentrations e. g., in average, at 0.26 mg PCBs. kg dw$^{-1}$, and at more than 940 ng. g dw$^{-1}$ for PAHs (Winiarski, 2014; Winiarski et al. 2006). These sediments and soils were also found contaminated by dioxins at about 36 ng. g dw$^{-1}$ (Winiarski, 2014), and by 4-nonylphenols and bisphenol A, at concentrations varying from 6 ng. g dw$^{-1}$ to 3400 ng. g dw$^{-1}$ (Wiest et al., 2018). However, MTE and non-polar PAHs found among SISs are unlikely to reach groundwaters. To illustrate, Pb and Cd were not recorded at depths below 1.5 m into the non-saturated zone of SISs (Winiarski et al. 2006). In contrast, polar organic pollutants were found to be transferred into aquifers as shown for some pesticides and pharmaceuticals (Pinasseau et al. 2019). These chemical contaminants represent potential energy- and carbon-sources for microorganisms, and can also be detrimental to the growth of some organisms. They can thus have significant impacts on the biology of the contaminated soils and sediments.

Functional inferences from the knowledge on bacterial genera suggested an occurrence of several aquifer taxa involved in the nitrogen and sulfur cycles but also in hydrocarbon degradation. *Campylobacter*, *Flavobacterium*, *Pseudomonas*, *Sulfurimonas* cells have been associated with nitrogen and sulfur respiration processes, and *Nitrospira* and *Leptospirillum* with nitrification. The oligotrophic nature of the aquifer waters (concentrations of biodegradable dissolved organic carbon < 0.5 mg. L$^{-1}$, Mermillod-Blondin et al., 2015) is thus likely to have induced a significant selective sorting of microbial taxa among the merged community. Most abundant aboveground taxa often require high energy (organic carbon) and nutrient levels to proliferate (Cho and Kim, 2000; Griebler and Lueders, 2009). Twice as much dissolved organic carbons were detected among aquifer waters of the experimental site recovered downstream the SIS (1.93 mg. L$^{-1}$ $\pm$ 0.77) than upstream (0.88 mg. L$^{-1}$ $\pm$ 0.27) (Mermillod-Blondin et al., 2015), and this effect was confirmed for other SIS (Mermillod-Blondin et al., 2015; Winiarski, 2014). Similarly, a large part of the bacterial taxa identified from aquifer biofilms was attributed to aboveground sources by the SourceTracker approach. Indeed, watershed runoff waters and detention basin deposits were found to have significantly contributed to the build-up of the observed biofilm community structures. These biofilms showed a high content of 16S rRNA gene sequences belonging to the *β-* and *γ-proteobacteria.*

According to the ecological concept of r/K selection, these *Proteobacteria* are often considered as r-strategists,
able to respond quickly to environmental fluctuations, and colonize more efficiently newly exposed surfaces than
other groups of bacteria (Araya et al., 2003; Fierer et al., 2007; Lladó and Baldrian, 2017; Manz et al., 1999;
Pohlon et al., 2010). Moreover, because they tend to concentrate nutrients (Flemming et al., 2016), biofilms are
likely to favor the survival of such opportunistic bacterial cells capable of exploiting spatially and temporally
variable carbon and nutrient sources. Here, taxa recovered from aquifer biofilms were previously recorded to have
the ability to use hydrocarbons as carbon- and energy sources e. g., *Nocardia*, *Pseudomonas*, *Sphingobium*, and
*Novosphingobium*. As indicated above, SIS and urban runoffs are well known to be highly polluted by such
molecules (e. g., Winiarski, 2014; Marti et al., 2017; Wiest et al., 2018). The r/K selection ecological concept thus
seems to apply to the biofilm community assemblages observed in this work.
Taxonomic allocations of the 16S rRNA OTUs suggested that the aquifer waters and biofilms likely harbored
opportunistic human, plant and animal pathogens of the genus *Finegoldia*, *Campylobacter*, *Haemophilus*,
*Duganella*, *Massilia*, *Nocardia*, *Aquabacterium*, *Flavobacterium*, *Pseudomonas, Streptococcus,* and *Aeromonas*.
A striking observation was the enrichment of 16S rRNA gene reads allocated to the *Nocardia* (about 4% of the
total reads) and *Pseudomonas* (about 35% of the total reads) in the biofilms recovered from clay beads incubated
downstream of the SIS. *Nocardia* and *Pseudomonas* 16S rRNA gene sequences were in much lower relative
proportions in the aboveground compartments. The genus *Pseudomonas* was previously found to be abundant
under low flow conditions and was often associated with biofilm formation (Douterelo et al., 2013). Moreover,
Pseudomonads are well-known for their ability at using hydrocarbons as energy and C-sources. Regarding the
*Nocardia* cells, there is a poor knowledge of their ecology, but a few reports indicated a tropism for hydrocarbon
polluted urban soils and sediments (e. g., Bernardin-Souibgui et al., 2018; Sébastian et al., 2014). There was no
additional approach to further investigate the molecular ecology of *Nocardia* cells found among the investigated
urban watershed. However, a *tpm* meta-barcoding analytical scheme could be applied on DNA extracts to further
explore the taxonomic allocations of the Pseudomonads and some other *tpm*-harboring genera. The applied *tpm*
meta-barcoding approach allowed an investigation of the coalescence of about 90 species among the investigated
watershed including 50 species of *Pseudomonas*, 11 species allocated to the *Aeromonas*, and some additional
species allocated to the *Nitrospira*, *Nitrosomonas*, *Stenotrophomonas*, *Xanthobacter*, and *Xanthomonas.* A single
*Aeromonas* species, *A. caviae,* was recorded among the above- and under-ground environments. More than 10
*Pseudomonas* species thriving in the recharged aquifer were detected among the aboveground compartments. *P.*
*umsongensis* and *P. chengduensis tpm* OTUs were detected aboveground, and represented a significant fraction of
the *tpm*-harboring bacteria retrieved from the aquifer samples. These two species were initially isolated from farm
soil and landfill leachates (Kwon et al., 2003; Tao et al., 2014), further supporting the hypothesis that such soil-
associated bacteria can be transferred from runoffs and urban sediments down to natural hydrosystems, and can
merge with aquifer communities. Regarding the *Pseudomonas* species that may pose health threats to humans, a
*tpm* OTU affiliated to *P. aeruginosa* was found to be shared between the surface compartments and the biofilm
*tpm* community developing on clay beads incubated downstream the SIS. *P. aeruginosa* thus had the properties
allowing an opportunistic development among the aquifer. This species is known for its metabolic versatility and
ability to thrive on hydrocarbons. This is an example of bacterial r-strategist being able to get established
opportunistically in aquifer biofilm communities impacted by urban pollutants. Apart from *P. aeruginosa,* the
species *P. putida* and *P. stutzeri,* frequently detected in soils and wastewater treatment plants (*e.g.* Igbinosa et al.,

2012; Luczkiewicz et al., 2015; Miyahara et al., 2010), were also recovered along the watershed and the aquifer. Although these two species were identified in human infections (Fernández et al., 2015; Noble and Overman, 1994), information about their virulence remains scarce. These species are therefore considered to be of less concern than *P. aeruginosa* and *A. caviae*, another opportunistic infectious agent (Antonelli et al., 2016) found in the aquifer. *P. putida* isolates have been shown to be involved in hydrocarbon degradation, and *P. stutzeri* can play part in the N-cycle either through denitrification or nitrogen-fixation.

## 5 Conclusions

The knowledge gained from the present study demonstrated that coalescence of microbial communities from an urban watershed with those of an aquifer can occur and yield novel assemblages. Specialized bacterial communities of aquifer waters were slightly re-shuffled by the aboveground communities. However, the assemblages observed among recent aquifer biofilms were found to be largely colonized by opportunistic r-strategists coming from aboveground compartments, and often associated with the ability to degrade hydrocarbons e. g., the Pseudomonads, *Nocardia* and *Novosphingobium* cells. The aquifer of the investigated site was found, for the first time, to be specifically colonized by species like *P. jessenii*, *P. chlororaphis*, and *P. resinovorans* but also undesirable human opportunistic pathogens such as *P. aeruginosa* and *A. caviae*. Artificial clay beads incubated in the aquifer through piezometers appeared highly efficient trapping systems (termed "germcatchers") to evaluate the ability of a SIS at preventing transfer of undesirable r-strategists to an aquifer. Nevertheless, the long term incidence of allochthonous bacteria on the integrity of aquifer microbiota remains to be investigated.

Free-living aquifer bacterial communities are not likely to be much impaired by exogenous cells. However, microbial communities developing as biofilms on inert surfaces might be significantly re-shuffled through selective sorting likely induced, in part, by aboveground chemical pollutants. Microbial biofilms are key structures in the transformation processes of several chemicals and nutrients. They often display much higher cell densities than free-living populations (Crump and Baross, 1996; Crump et al., 1998; van Loosdrecht et al., 1990). Here, we have demonstrated that runoff and SIS bacterial taxa can colonize solid matrices of a deep aquifer. These modified communities could (i) alter geochemical processes which can indirectly impact other groundwater inhabitants e. g., the amphipod *Niphargus rhenorhodanensis* and other taxa presented in Foulquier et al. (2011), or (ii) directly impact these inhabitants by inducing a modification of their microbial contents, and potentially of their behavior. The stygofauna feed on bacteria, and is well known to be significantly colonized by bacteria (e. g., Smith et al., 2016). The next step in these studies will be to investigate whether native aquifer biofilm communities can resist to repeated invasions by opportunistic r-strategists and if these allochthonous bacteria will impact the ecological health of the stygofauna.

*Data availability.* The 16S rRNA gene sequences are available at the European Nucleotide Archive (www.ebi.ac.uk/ena) using the following accession numbers: PRJEB33510 (IB), PRJEB21348 (DB), PRJEB29925 (AQ), and PRJEB33507 (WS), and the *tpm* gene sequences using the PRJEB33622 accession number.

*Supplement.* The supplementary materials related to this article is available online at:

556

*Author contribution.* BC coordinated and designed the experiments. YC, VRN, TW, FMB, RB, LM, RM, FV, EB, DB, JV, and BC performed the experiments and contributed at the analysis of the datasets. YC and BC prepared the manuscript with contributions from all co-authors.

560

*Competing interests.* The authors declare that they have no conflict of interest.

*Acknowledgments.* This work was partly funded by the French national research program for environmental and occupational health of ANSES under the terms of project "Iouqmer" EST 2016/1/120, l'Agence Nationale de la Recherche through the ANR-16-CE32-0006 (FROG) and ANR-17-CE04-0010 (Infiltron) projects, by Labex IMU (Intelligence des Mondes Urbains), the Greater-Lyon Urban Community, the School of Integrated Watershed Sciences H2O'LYON (ANR 17-EURE-0018), the MITI CNRS project named Urbamic, and the Urban School of Lyon (ANR-17-CONV-0004). Authors thank the OTHU network for technical assistance and financial supports, and the reviewers of this paper for their constructive comments.

Edited by:
Reviewed by:

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

**Table 1.** Coalescence of surface and aquifer bacterial communities inferred by the SourceTracker Bayesian approach and the 16S rRNA gene meta-barcoding dataset*

| samples | | WS | | DB | | IB | | AQ_wat_up | | unknown | |
|---|---|---|---|---|---|---|---|---|---|---|---|
| | | mean | rsd | mean | rsd | mean | rsd | mean | rsd | mean | rsd |
| 1 - waters | AQ_wat_dw1 | 0.3% | 33.3 | 0.3% | 43.3 | 7.5% | 42.6 | 19.7% | 30.6 | 72.3% | 4.8 |
| | AQ_wat_dw2 | 10.2% | 50.6 | 17.6% | 10.3 | 9.7% | 18.8 | 25.7% | 15.4 | 36.9% | 6.7 |
| | AQ_wat_dw3 | 5.0% | 9.0 | 5.0% | 29.1 | 3.8% | 32.0 | 70.7% | 1.9 | 15.5% | 2.3 |
| | AQ_bio_dw1 | 8.6% | 23.5 | 25.0% | 19.1 | 3.9% | 74.1 | 56.7% | 6.9 | 5.8% | 7.9 |
| | AQ_bio_dw2 | 13.6% | 28.0 | 28.4% | 14.1 | 2.9% | 46.0 | 48.2% | 6.52 | 6.8% | 11.8 |
| 1 - biofilms | AQ_bio_dw3 | 3.4% | 17.1 | 13.9% | 18.4 | 5.5% | 39.3 | 72.1% | 1.85 | 5.2% | 29.8 |
| 2 - biofilms | AQ_bio_up1 | 32.2% | 14.5 | ✕ | ✕ | ✕ | ✕ | 61.3% | 9.5 | 6.8% | 23.7 |
| | AQ_bio_up2 | 56.6% | 12.6 | | | | | 36.4% | 18.3 | 7.0% | 15.7 |
| | AQ_bio_up3 | 44.0% | 6.6 | | | | | 48.1% | 8.1 | 7.8% | 10.8 |

* Two analyses are shown from sub-sampled datasets set at 1000 reads: (1) reads from WS, DB, IB, and aquifer waters from upstream the SIS were considered as the sources of taxa for the aquifer samples downstream the SIS; (2) reads from WS and the aquifer waters upstream the SIS were considered as the sources of taxa for the aquifer biofilms recovered upstream the SIS. SourceTracker was run 3 times using the 16S rRNA gene OTU contingency table and the default parameters. Relative contributions of the sources were averaged. Relative standard deviations (%RSD) are indicated, and used as confidence values. RSD > 100% indicates low confidence on the estimated value. WS: Watershed runoff waters; DB: Detention basin sediments; IB: Infiltration basin sediments. Sequences that could not be attributed to one of the tested sources were grouped under the term unknown.

**Table 2.** Aquifer 16S rRNA gene (*rrs*) OTUs detected in the upper compartments of the investigated watershed and SIS*.

| | Upstream SIS | | | | | |
|---|---|---|---|---|---|---|
| | AQ_bio_up1 | AQ_bio_up2 | AQ_bio_up3 | AQ_wat_up1 | AQ_wat_up2 | AQ_wat_up3 |
| **(A) Number of aquifer *rrs* OTUs shared with the upper compartments** | 185/220 | 110/160 | 118/173 | 93/143 | 80/164 | 165/464 |
| **(B) Relative abundance of the shared *rrs* OTUs in the aquifer (in %)** | 99.4 | 95.0 | 96.4 | 43.8 | 45.4 | 33.6 |
| **(C) Relative abundance of the shared *rrs* OTUs in the upper compartments (in %)** | 24.9 | 15.5 | 15.8 | 9.7 | 9.8 | 11.3 |
| | Downstream SIS | | | | | |
| | AQ_bio_dw1 | AQ_bio_dw2 | AQ_bio_dw3 | AQ_wat_dw1 | AQ_wat_dw2 | AQ_wat_dw3 |
| **(A) Number of aquifer *rrs* OTUs shared with the upper compartments** | 340/403 | 308/353 | 321/362 | 203/523 | 357/594 | 468/1052 |
| **(B) Relative abundance of the shared *rrs* OTUs in the aquifer (in %)** | 99.4 | 99.4 | 99.6 | 52.2 | 83.4 | 53.7 |
| **(C) Relative abundance of the shared *rrs* OTUs in the upper compartments (in %)** | 29.7 | 30.7 | 39.4 | 12.5 | 32.0 | 24.2 |

*the number or relative number of shared aquifer *rrs* OTUs found in the upper compartments (WS, DB, IB) was computed per aquifer sample recovered upstream (up) or downstream (dw) the SIS (see Fig. 1 for the sampling design), after a re-sampling of the reads set at 20,624 per sample. AQ_wat: Aquifer waters; AQ_bio: Aquifer clay beads biofilms; up: upstream the SIS, dw: downstream the SIS.

**Table 3.** Relative distribution of *tpm* reads per OTU (mean ± sd) shared between the upper compartments and the aquifer, and that were allocated to well-defined species.[1]

| Genus | Species | OTU code[2] | WS | DB | IB | AQ_Wat_up | AQ_Bio_up | AQ_Wat_dw | AQ_Bio_dw |
|---|---|---|---|---|---|---|---|---|---|
| *Nitrosomonas* | *oligotropha* | Otu00035 | nd | 1.5 ± 3.40 | 0.15 ± 0.30 | nd | + | + | nd |
| *Pseudomonas* | *aeruginosa* | Otu00066 | 0.42 ± 1.13 | nd | + | nd | nd | nd | 0.17 ± 0.30 |
| *Pseudomonas* | *chengduensis* | Otu00024 | nd | + | + | 20.43 ± 35.39 | nd | + | nd |
| *Pseudomonas* | *extremaustralis* | Otu04178 | nd | + | nd | nd | nd | + | + |
| *Pseudomonas* | *fragi* | Otu00197 | 0.61 ± 4.05 | nd | nd | nd | nd | + | nd |
| *Pseudomonas* | *pseudoalcaligenes* | Otu00197 | 0.07 ± 0.38 | + | nd | + | nd | nd | nd |
| *Pseudomonas* | *putida* | Otu00800 | + | + | nd | nd | nd | + | nd |
| *Pseudomonas* | *stutzeri* | Otu00119 & Otu10066 | 0.06 ± 0.33 | nd | + | 3.06 ± 5.29 | nd | nd | + |
| *Pseudomonas* | *umsongensis* | Otu00005 | + | + | nd | 0.41 ± 0.71 | 17.79 ± 20.11 | 5.34 ± 8.58 | 11.71 ± 13.17 |
| *Pseudomonas* | *viridiflava* | Otu00204 | 0.06 ± 0.31 | nd | 0.3 ± 1.09 | nd | nd | 0.07 ± 0.12 | nd |
| *Stenotrophomonas* | *acidaminiphila* | Otu00072 & Otu01119 | 0.09 ± 0.42 | 0.29 ± 0.91 | 0.06 ± 0.22 | nd | nd | + | nd |
| *Xanthobacter* | *autotrophicus* | Otu00501 | + | + | nd | nd | nd | 0.06 ± 0.11 | + |
| *Xanthomonas* | *axonopodis/campestris* | Otu00019 & Otu00878 | 0.25 ± 0.75 | nd | 1.24 ± 2.07 | 16.04 ± 27.78 | nd | nd | + |
| *Xanthomonas* | *cannabis* | Otu00004 | 3.74 ± 9.47 | nd | nd | nd | + | + | + |

[1] All reads from *tpm* OTUs shared between the upper compartments and the aquifer were used to compute the relative abundances.
[2] *tpm* sequences of the OTUs are shown in Table S8. WS: Watershed runoff waters; DB: Detention basin deposits; IB: soil of the infiltration basin; AQ_water: Aquifer waters; AQ_bio: Aquifer biofilms. + : OTUs with a relative abundance < 0.05%. nd : not detected.

**Figure captions**

**Figure 1.** Scheme illustrating the trajectory of urban runoffs from the industrial watershed (WS) towards the stormwater infiltration system (SIS) investigated in this study. The urban watershed is located in Chassieu (France). The SIS is made of a detention basin (DB) and an infiltration basin (IB), and is connected to the Lyon 200 km$^2$ east aquifer (AQ).

**Figure 2.** General features of the V5-V6 16S rRNA gene meta-barcoding DNA sequences obtained from runoff, SIS, and aquifer samples. See Figure 1 for a description of the experimental design. Panel (A) illustrates the relative abundance of the main bacterial phyla observed per compartment, and (B) shows boxplots illustrating the variations observed per compartment between the Shannon diversity indices computed from the V5-V6 16S rRNA gene OTU contingency table (sub-sampled dataset). One-way ANOVA with multiple Tukey post hoc tests were performed to investigate the differences between compartments. Different letter codes indicate significant differences ($p<0.05$). WS, runoff waters from the watershed; DB: sediments from the detention basin; IB: soils from the infiltration basin; AQ_water: Aquifer waters; AQ_bio: Aquifer clay beads biofilms.

**Figure 3.** PCoA analysis of weighted UniFrac dissimilarities between the V5-V6 16S rRNA gene OTU profiles of the watershed runoff waters (WS), urban sediments and soils from the connected detention (DB) and infiltration (IB) basins, and waters (AQ_water) and biofilms (AQ_bio) from the connected aquifer. See Figure 1 for a description of the experimental design. Ellipses are representative of the variance observed (standard error) between the ordinations of a group of samples. PERMANOVA tests confirmed the significance ($p < 0.001$) of the groupings. The proportion of the eigenvalue per axis over the sum of all eigenvalues was 25.0% for PCoA1 and 17.6% for PCoA2. The sub-sampled V5-V6 16S rRNA gene dataset was used for the computations.

**Figure 4.** Relative numbers of potentially pathogenic bacterial genera along the watershed. The abundance (rel. abund.) of bacterial genera exclusively detected in upper compartments (A) or both in the upper compartments and aquifer (B) are presented. Size of bubbles is proportional to the relative abundance (in %) of each bacterial genus per sampled compartment. WS: runoff waters from the watershed; DB: sediments from the detention basin; IB: sediments from the infiltration basin; AQ_water: Aquifer waters; AQ_bio: Aquifer clay beads biofilms.

Fig. 1 - Colin et al. hess-2020-39

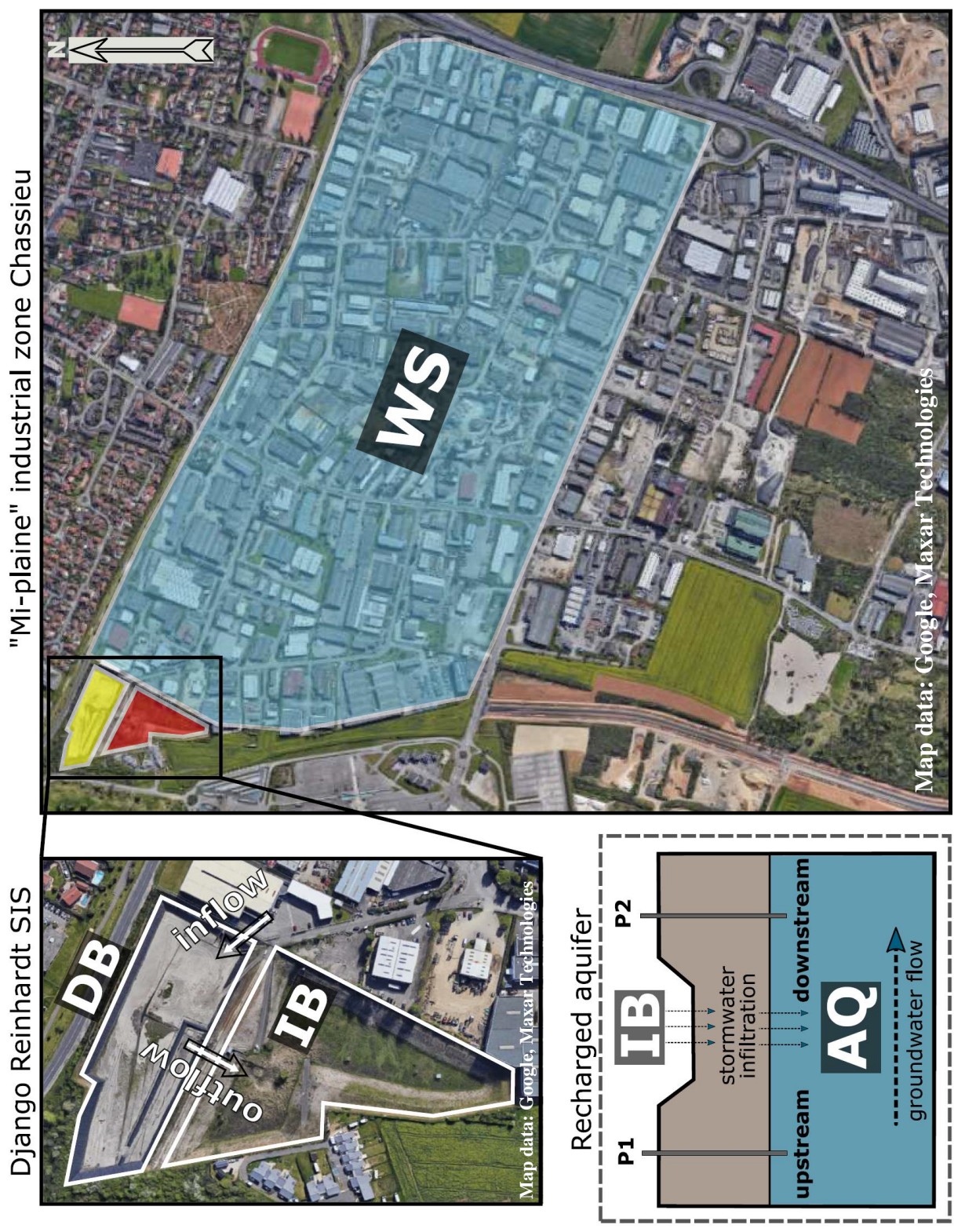

852

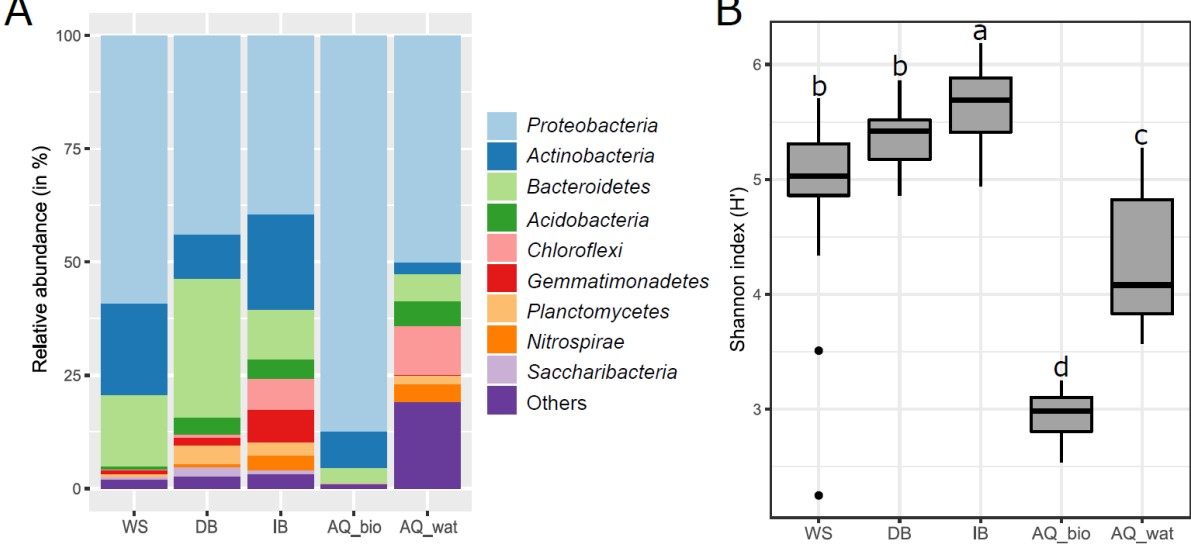

853

    Fig. 3 - Colin et al. hess-2020-39

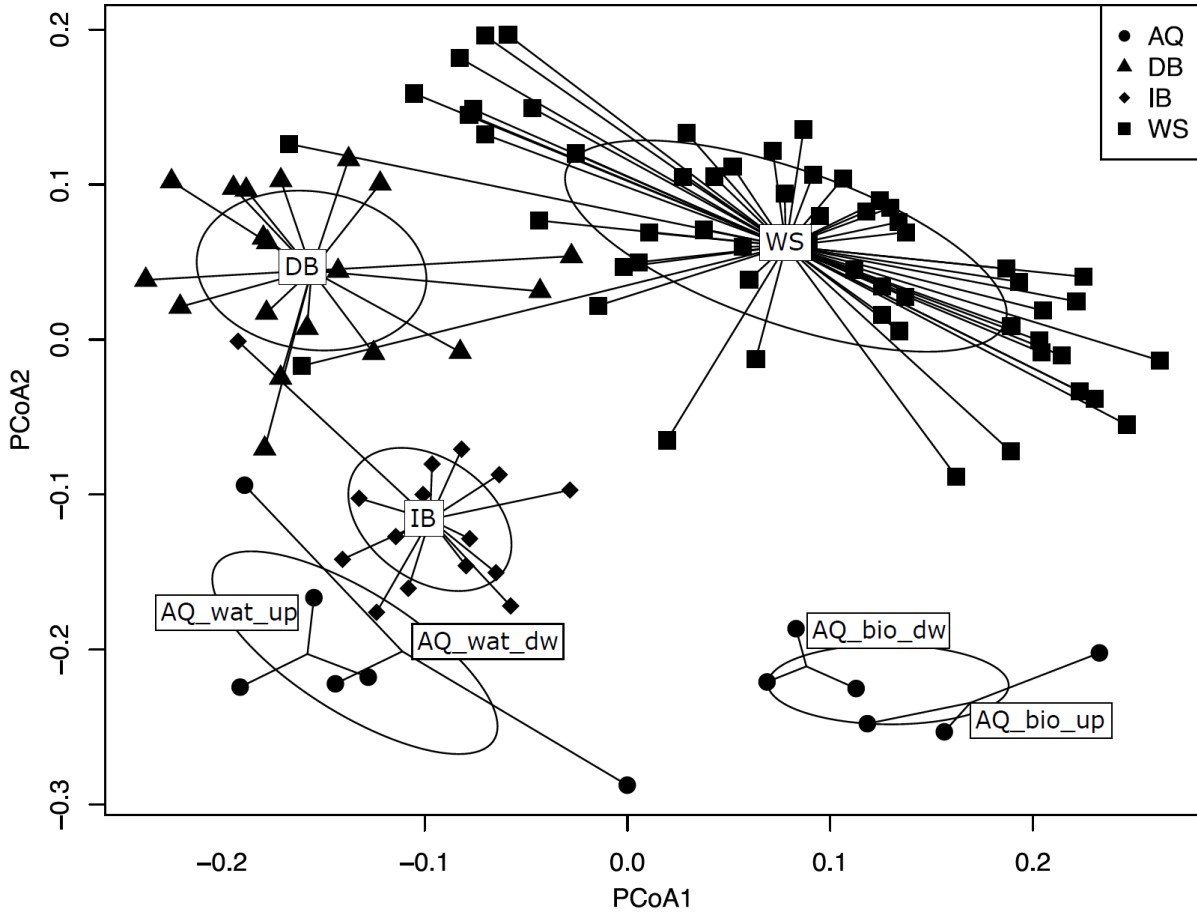

 Fig. 4 - Colin et al. hess-2020-39

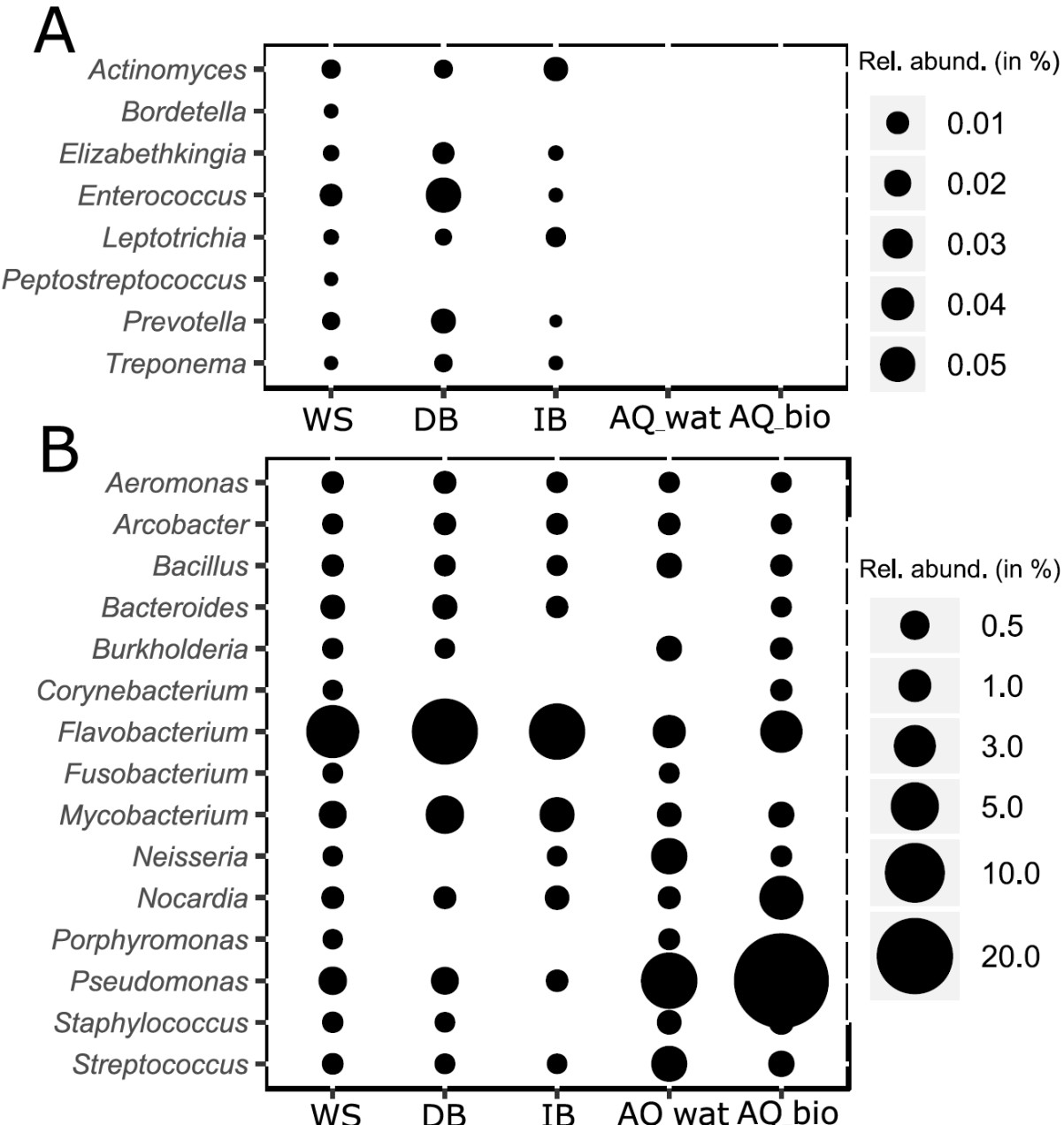