# Peer review of "Coalescence of bacterial groups originating from urban runoffs 1 and artificial infiltration systems among aquifer microbiomes 2 3 Yannick Colin1‡, Rayan Bouchali1, Laurence Marjolet1, Romain Marti1, Florian Vautrin1,2, Jérémy Voisin1,"

_Hydrology and Earth System Sciences, 2020_

## Referee Comment (RC1) · Louis Carles (Referee) · 13 Mar 2020

General comments The present work describes the influence of urban runoffs and artificial infiltration systems on aquifer microbioms. The manuscript is well written, easily readable and well structured. I have few general comments (open questions) and some minor suggestions. In what extend the comparison of samples taken at different time is meaningful? How did you integrate the time-scale in your analyses and discussion? In the discussion section, the authors explain the shifts observed in microbial communities structure by emitting hypotheses linked to nutrient availability and environmental factors. However, physical-chemical data describing the different

compartment of the studied site are lacking. What would be the effect of the species coming from surface environments on the functioning of the microbial communities in the aquifer? Specific comments L91-94. I agree with this sentence. However, one should also take into account the fact that the transition between free-living organisms and biofilm is taxa-dependant. Meaning that the biofilm is not 100% reflecting the diversity of free-living microorganisms living in the aquifer. This is confirmed by H' values (AQ_bio < AQ_wat) in Figure 2B. L204-205. How could you explain the higher diversity in IB compared to WS and DB? Figure 3. It would be useful to add the eigenvalues for both axes. L213-215 and Figure S2. Dissimilarities instead of similarities? Technical corrections L226. Remove "(Table 2)". Table 1 should be Table 2, and inversely. Please make the correction in the text, accordingly. L305. Add "species" after "Some".

Please also note the supplement to this comment:
https://www.hydrol-earth-syst-sci-discuss.net/hess-2020-39/hess-2020-39-RC1-supplement.pdf

---

## Referee Comment (RC2) · Anonymous Referee #2 · 16 Mar 2020

The article entitled "Coalescence of bacterial groups originating from urban runoffs and artificial infiltration systems among aquifer microbiomes" presents the results of a study monitoring the microbial community assemblage though a stormwater infiltration system. The main findings of the study surrounded the contribution of each of the sources (the runoff and natural aquifer) to the combined system. An additional tracking of the biodegradation-related tpm gene allowed for finer species level resolution of the structure and shifts in the microbial community. As stormwater infiltration systems are becoming more commonly applied, readers will be interested in understanding their community assemblage.

[Figure]

However, the study requires additional information and revisions to clarify the presented investigation. Recommended edits and improvements to enhance the presented study are presented within three categories: Major, Minor, and Grammar revisions. The nine major revisions recommended in this review emphasize required improvements on the presentation of the analytical methods, bioinformatic approaches, and streamlining the discussion. The minor and grammar portions present less critical revisions.

Major

Maj1. The presentation of the sequencing process employed is inadequate. The current text highlights that the sequences were run on a Illumina MiSeq, without providing additional details. First, the study does not mention how the nucleic acids are extracted from the samples, checked for quality, stored, and shipped to the facility. These points must be clarified. Second, the study must clarify within section 2.2 several key points with respect to the sequencing protocol: (1) a citation for the primers used to target the 16S rRNA gene, (2) the protocol followed by the laboratory must be unambiguously indicated or referenced (TruSeq, Nextera, etc.), (3) the target length of the sequences, and (4) whether the sequence reads were paired-end or single. The current presentation does not follow the MINSEQ guidelines for the required information for sequencing studies. Third, the presented study does not mention either positive mock community or negative comparison controls (and how those samples are incorporated into the analyses to remove contaminating sequences). The authors must present these controls.

Maj2. The results of the sequencing campaign additionally requires a more comprehensive presentation. L193-194 presents the total sequencing reads, but must present the average and range of reads per sample. A supplemental table must be provided with the raw and processed sequencing counts for each sample.

Maj3. Additionally, to explore quantitatively the mixing ratios and why certain communities are providing more biomass, the actual concentration of the community within

these compartments should be mentioned or addressed as to why these measurements were neglected.

Maj4. The bioinformatic processing pipeline requires additional information. First, the approach presented divides the 16S rRNA amplicons into 97% OTUs. However, current best practices recommends utilizing the amplicon sequencing variants (ASV) approach (Knight et al., 2018). The authors should either update their approach to the ASV methodology or provide a concise defense as to why they selected the OTU approach. Second, a rarefaction analysis is presented to subsample the dataset at 20,624 sequences. This approach has been recently called into question for more directed comparisons (McMurdie and Holmes 2014). The authors should present a concise defense as to why rarefaction was employed. To bolster this defense, Figure S1 should display the rarefaction curve for the raw data, not the previously subsampled 20,624 dataset (this comment connects with Maj2 in the need to present additional information).

Knight, R., Vrbanac, A., Taylor, B. C., Aksenov, A., Callewaert, C., Debelius, J., ... & Melnik, A. V. (2018). Best practices for analysing microbiomes. Nature Reviews Microbiology, 16(7), 410-422.

McMurdie, P. J., & Holmes, S. (2014). Waste not, want not: why rarefying microbiome data is inadmissible. PLoS computational biology, 10(4).

Maj5. In the SourceTracker default code, the rarefied sample is then rarefied further to 1000. This procedure should be repeated to draw those 1000 reads from the full dataset, not the previously rarefied data.

Maj6. L319-337 presents a great overview of the study that is more appropriate for the abstract rather than the discussion. This section should be removed in its entirety.

Maj7. Throughout the text, the presence of a specific 16S rRNA transcript often is utilized to state the presence of a specific function within the community, notably within

the abstract (e.g., L25, L27). Whereas the 16S taxonomical assignment is a good indicator that a specific function is likely encoded on the metagenome of the community, the linkage is not directly shown through the 16S survey and must be caveated by "likely", "putative", or "predicted to be". This is recognized more consistently within the discussion of the results, but must be maintained throughout the text to recognize that the assignment provided by FAPROTAX is a hypothesis.

Maj8. The authors commendably provided the raw data as publicly available datasets through EBI. Additionally, the authors should provide all code utilized to process these data as a part of the supplemental materials to allow future readers to reconstruct the presented results.

Maj9. The authors are encouraged to focus on improving the English language and grammar associated with the presented article. A non-exhaustive list of suggested grammar improvements is provided in the final section of this review, but additional editing services are recommended to enhance the clarity and accuracy of the text.

Minor

Min1. The bulk physical and chemical properties of the sampling sites should be presented or directly cited such as pH, temperature, electroconductivity etc. Additionally, please replace "for which physico-chemical and biological monitorings have been implemented" with "that records both physico-chemical and biological properties."

Min2. L34 – Please clarify what is meant by "DNA imprints allocated"

Min3. L70-75 – Please provide citations in support of these claims.

Min4. L78-L79 – Replace "The tested hypotheses were that" with "Two hypotheses were tested:". Because these statements are presenting the underlying hypotheses (supported or rejected), all qualifiers for the verbs must be removed. Therefore, remove L78 "should" and L79 "could also". L79 – Replace "but" with ", and". Similarly with L88-90, please replace "was likely to be" with "will be"

[Figure]

Min5. L291-307 – The long list of species mapped to the Pseudomonas genera is difficult to interpret in the currently presented form. Please condense this section for readability.

Min6. Throughout the text, ensure that a comma appears after Latin abbreviations such as i.e., and e.g.,

Min7. Figure 1, please italicize the names of the phyla.

Grammar

L23 – Please add "basins" after "detention". Currently, this sentence presents a broken list of items.

L24 – Please replace "made up" with "comprised"

L27 – Please add a comma before "but a higher"

L28 – Please replace "a tracking" with "the tracking"

L29 – Please replace "including the" with "including", remove "among these communities", and replace "the Pseudomonas" with "Pseudomonas"

L31 – Please replace "respectively, of" with "respectively, in"

L32 – Please remove the comma before ", and waters" and add "to be" after "found"

L34 – Please add a comma before "but only"

L36 – Please add "in" after "than"

L48 – Please replace "during" with "over"

L53 – Please replace "towards" with "capturing" and remove the comma after "metals)"

L59-60 – Please add "both the" before "water transit", replace ", but also the biology" with " and biological properties", replace "cover and root" with "cover, root", replace "worms" with "worm population", and add "composition" after microbiota.

L76 – Please replace "Here, the" with "This", replace "explored" with "explores", and remove the commas around ", with a thick vadose zone (> 10 m),"

L83 – Please replace "It" with "The site"

L84-85 – Please replace "It has an average vadose thickness of" with "The average vadose thickness of the site is"

L86-88 – Please remove "large", replace "built" with "recorded", remove ", in order", add "the" before "bacterial community", add "the" before "top", replace "among" with "into", and add "the" before "biofilm".

L92 – Please replace "among" with "within" and replace "while" with ", whereas"

L99 – Please remove "To go deeper into these inferences," and replace "were built" with "were then assembled"

L101 – Please remove the comma after "level," and replace ", and allowed gaining further insights on" with "to explore with a higher resolution"

L103 – Please replace "with" with "within"

L108 – Please replace "about" with "approximately"

L110 – Please add "a" before "part"

L112 – Please remove the comma before ", built"

L113-114 – Please replace "development of a plant cover" with "plant cover development

L116 – Please replace "deeply" with "previously" and check the format requirements for citations.

L118 – Please replace "were considered for this" with "are investigated within"

L120-122 – Please replace "have been" with "were", add an "a" before "50", and remove

"of the DB"

L122 – Please remove the comma before the "and"

L124 – Please replace "had been" with "were"

L126 – Please remove the comma before "at a"

L127 – Please replace "have been" with "were"

L130 – Please replace "had been" with "was"

L132 – Please add "subsequently" before "discarded"

L133 – Please replace "using clay" with "from clay"

L134 – Please replace "the same piezometers as those for the aquifer water samplings" with "the piezometers described above" and delete the subsequent sentence whereas including the citation just after (n = 6 samples).

L137 – Please revise this title to be more informative, such as "Generation and sequencing of the DNA amplicons"

L139 – Please replace "with Illumina MiSeq technology" with "on an Illumina MiSeq"

L165 – Please capitalize "BLAST"

L166 – Please remove "in order"

L168 – Please replace "carried out" with "performed"

L178 – Please remove "down"

L179 – Please remove ", in order"

L185 – Pease remove the comma before ", with"

L199 – Please replace "superior" with "greater than"

L205 – Please replace "of detention" with "withdrawn from the detention"

L207 – Please add a "the" before "clay" and replace "for" with "over"

L212 – Please replace of "to be made of" with "to contain"

L213 – Please replace "a same" with "the same"

L214 – Please replace "while" with "whereas the"

L216 – Please replace "found" with "to be found"

L226 – Please replace "while" with "whereas"

L233 – Please replace "much to" with "substantially to the" and remove "Content of"

L236 – Please replace "even though" with "although"

L238 – Please remove "In order"

L244 – Please add "and" before "Nitratireductor."

L245 – Please replace "while" with ", and"

L260 – Please replace "the SIS" with "of the SIS"

L261 – Please add "and" before "Flavobacterium"

L262 – Please replace "while" with "whereas"

L264 – Please add "and" before "Meganema"

L265 – Please replace "found: with "found to be"

L276 – Please replace "It is to be noted that" with "Notably,"

L277 – Please replace "part" with "representative"

L280 – Please replace "deeper" with "further"

L287 – Please remove "the"; additionally, because the data is already given as Table

S6, I recommend removing the exact percentages from this paragraph.

L295 – Please add a comma before "but"

L296 – Please add a comma before "but"

L316 – Please replace "while" with "when"

L345 – Please remove the comma before ", and"

L347 – Please replace "that can also enhance" with "enhances"

L348 – Please remove "Nevertheless" and replace "has induced" with "induces"

L350 – Please replace "that" with "than"

L352 – Please replace "the SIS" with "of the SIS"

L353 – Please replace "the SIS" with "of the SIS"

L360 – Please replace "the SIS" with "of the SIS"

L385 – Please add a comma before ", and significant"

L389 – Please replace "while" with "whereas"

L391 – Please replace "to likely" with "likely"

L396 – Please replace "the SIS" with "of the SIS"

L398 – Please remove the comma before ", and was"

L400 – Please add a comma before "but a few"

L404 – Please replace "in order to go deeper into" with "to explore further"

L414 – Please remove the comma before ", and can"

L418 – Please replace "It would thus be part of the r-strategists that could get opportunistically established" with "Therefore, an r-strategist would likely establish opportunistically"

L422 – Please remove "However,"

L425 – Please add "to be" after "shown"

L429 – Please remove the comma before ", and yield"

L431 - Please add "to be" after "found"

L432 – Please replace "at degrading" with "to degrade"

L436 – Please use an alternative term to "germcatchers"

L437 – Please remove "down"

L439-440 – Please improve the wording of the sentence beginning with "Free-living"

L443 – Please remove "to these"

L687 – Please replace "runoffs" with "runoff"

L688-689 – Please remove the commas after "(A)," and "(B),"

L700 – Remove "down the aquifer"

Table 1. Capitalize "Downstream SIS"; in the caption, replace "in (A),", "in (B)," and "in (C)," with "(A)", "(B)", and "(C)", respectively. Remove all occurrences of "is indicated"

[Figure]

---

## Author Comment (AC1) · 14 Apr 2020

Replies (Rep1) to reviewer # 1 (R1-Question#) (Louis Carles, 13 Mar 2020) (line numbers are those of the initial submission)

R1-Q1: In what extend the comparison of samples taken at different time is meaningful? How did you integrate the time-scale in your analyses and discussion?

Rep1-Q1: Here, the hypothesis was that urban aquifers fed by SIS should harbor microbiota reflecting the coalescence (community assemblages and selective sorting) of aboveground microbial communities with those of the aquifer. This hypothesis was tested by comparing representative bacterial taxonomic units defined by two genetic markers (16S rRNA and tpm genes) and observed among significant compartments of the watershed (urban runoffs, sediments / urban deposits, soils in contact with the urban runoffs), with those from aquifer samples (water and biofilms). Time scale was considered in previous reports such as Voisin et al. 2018 (doi : 10.1016/j.scitotenv.2018.05.094) and Marti et al. 2017 (10.1038/s41598-017-13594-8). From these latter analyses, core and versatile bacterial communities could be observed among the investigated system. Stochasticity was not the leading rule among the investigated system because of strong environmental constraints and the significant occurrence of particular C-sources such as polycyclic aromatic hydrocarbons (PAH) e. g. see Fig. 3 which shows a clear discrimination of community profiles according to the investigated compartment. It is to be noted that over the investigated time period, no significant urbanistic changes occurred over the observatory site. The Django-R urban watershed is part of the OTHU (Lyon Urban Waters Field Observatory) long term experimental site dedicated to urban waters (http://www.graie.org/othu/)); therefore, changes over the watershed are tracked in collaboration with the Lyon Metropolitan water services. This last point is now indicated in the Materials and Methods section.

From line 109, the following sentence was added: ". . . No significant urbanistic changes impacting the urban watershed were recorded during the investigated period. . .."

R1-Q2 : In the discussion section, the authors explain the shifts observed in microbial communities structure by emitting hypotheses linked to nutrient availability and environmental factors. However, physical-chemical data describing the different compartment of the studied site are lacking.

Rep1-Q2: These datasets were cited in the discussion and not described in the result section because they were published elsewhere. In fact, we've published several physical-chemical datasets in the frame of this collaborative long-term investigation which is addressing several issues related to urban waters and their management.

To clarify these issues, the following sentences were added in the discussion: From line 365: "..In fact, chemical pollutants have been shown to be significantly washed-off or transported with particles during rain events e. g. El-Mufleh et al. (2014; doi.org/10.1007/s11356-013-2490-3), and some of these were found to reach aquifers fed by SISs (e. g. Pinasseau et al. 2019; doi.org/10.1016/j.scitotenv.2019.03.489). Among these pollutants, Bernardin-Souibgui et al. (2018) reported that urban sediments found in the detention basin of the experimental site were heavily polluted by polycyclic aromatic hydrocarbons (PAH). Their contents were estimated at 197±36 ng/g dw (dry weight) for light PAHs, and at 955±192 ng/g dw for heavy PAHs. PCBs were also recorded for the seven congeners of the European norm for a total of 0.2 to 2.1 mg/ kg dw (Sebastian et al., 2014). Metallic trace elements (MTE) were recorded in significant amounts, with Cu being found at about 280 mg / kg dw, Pb at about 200 mg / kg dw, Zn at about 1600 mg, and Cd at about 5 mg / kg dw (Sebastian et al., 2014). MTE, PCBs and PAHs were also recorded in the soils of the infiltration basin at similar ranges e. g. in average, at 0.26 mg PCBs / kg dw, and at more than 940 ng/g dw for PAHs (Winiarski, 2014; temis.documentation.developpement-durable.gouv.fr/docs/Temis/0081/Temis-0081850/21970_A.pdf; Winiarski et al. 2006; doi.org/10.1007/s10040-006-0073-9). These sediments and soils were also found contaminated by dioxins at about 36 ng / g dw (Winiarski, 2014), and by 4-nonylphenols and bisphenol A, at concentrations varying from 6 ng/g dw to 3400 ng/g dw (Wiest et al., 2018; doi.org/10.1016/j.envpol.2018.09.138). MTE and non-polar HAPs found among SISs are unlikely to reach groundwaters. To illustrate, Pb and Cd were not recorded at depths below 1.5 m into the non-saturated zone of SISs (Winiarski et al. 2006). In contrast, polar organic pollutants were found to be transferred into aquifers as shown for some pesticides and pharmaceuticals (Pinasseau et al. 2019). These chemical contaminants represent potential energy- and carbon-sources for micro-organisms, and can also be detrimental to the growth of some organisms. They can thus have significant impacts on the biology of the contaminated soils and sediments of SISs. Functional inferences from the . . ."

From line 372: "...Twice as much dissolved organic carbons were detected among aquifer waters of the experimental site downstream the SIS (1.93 mg/L ± 0.77) than upstream (0.88 mg/L ± 0.27) (Mermillod-Blondin et al., 2015; Winiarski, 2015)..."

R1-Q3: What would be the effect of the species coming from surface environments on the functioning of the microbial communities in the aquifer?

Rep1-Q3 : Consequences of these changes are multi-factorial.

From line 437, we've clarified these issues by modifying the text below – new sentences are in red.

"... The long term incidence of allochthonous bacteria on the integrity of aquifer microbiota remains to be investigated. Free-living bacterial communities are not likely to be much impaired but those developing as biofilms on inert surfaces might be. Microbial biofilms are key structures in the transformation processes of several elements and nutrients. They often display much higher cell densities than free-living populations (Crump and Baross, 1996; Crump et al., 1998; van Loosdrecht et al., 1990). Here, we have demonstrated that runoff and SIS bacterial taxa can colonize solid matrices of a deep aquifer. These modified communities could (i) alter geochemical processes which can indirectly impact other groundwater inhabitants e. g. the amphipod Niphargus rhenorhodanensis and other taxa presented in Foulquier et al. (2011; doi.org/10.1007/s10021-011-9484-0), or (ii) directly impact these inhabitants by inducing a modification of their microbial contents, and potentially of their behavior. The stygofauna feed on bacteria, and is well known to be significantly colonized by bacteria (e. g. Smith et al., 2016; doi.10.1038/srep32738). The next step in these studies will be to investigate whether native aquifer biofilm communities can resist to repeated invasions by opportunistic r-strategists, and if these allochthonous bacteria will impact the ecological health of the stygofauna. "

R1-Q4 : L91-94. I agree with this sentence. However, one should also take into account the fact that the transition between free-living organisms and biofilm is taxadependant. Meaning that the biofilm is not 100% reflecting the diversity of free-living microorganisms living in the aquifer. This is confirmed by H' values (AQ_bio < AQ_wat) in Figure 2B.

Rep2-Q4 : OK, we've tried to clarify this issue by adding the following sentence: From L94: "Clay bead biofilms were found to capture the most abundant aquifer taxa, and taxa that could not be detected from grab samples. Although, it is to be noted that some taxa are not likely to efficiently colonize clay beads over short time periods."

R1-Q5 : L204-205. How could you explain the higher diversity in IB compared to WS and DB?

Rep2-Q5: we've added the following sentences to explain this difference. From L354: "...It is to be noted that soils of the infiltration basin showed higher bacterial diversity than those of the sediments of the detention basin and runoffs. This is most likely related to a development of plant-associated bacteria in this compartment. Indeed, the infiltration basin was covered by several plant species of Magnoliophyta like Rumex sp. which can disseminate rapidly through rhizomes (Bedell et al., 2013; doi.org/10.2166/wst.2013.526) and generate multiple ecological niches for bacteria."

R1-Q6 : Figure 3. It would be useful to add the eigenvalues for both axes.

Rep2-Q6: The proportion of the eigenvalue to the sum of all eigenvalues was calculated in order to estimate the importance of the axes in discriminating the samples from each other (PCoA1: 25.0% and PCoA2: 17.6%). These values will be indicated.

From L699, the following sentence was added: "The proportion of the eigenvalue per axis over the sum of all eigenvalues was 25.0% for PCoA1 and 17.6% for PCoA2."

R1-Q7 : L213-215 and Figure S2. Dissimilarities instead of similarities?

Rep2-Q7: we've computed similarities (100 - % of dissimilarities) to facilitate the appreciation of differences between samples

R1-Q8 :Technical corrections L226. Remove "(Table 2)". Table 1 should be Table 2, and inversely. Please make the correction in the text, accordingly. L305. Add "species" after "Some". Rep2-Q8: all these problems were considered and fixed

Please also note the supplement to this comment:
https://www.hydrol-earth-syst-sci-discuss.net/hess-2020-39/hess-2020-39-AC1-supplement.pdf

---

## Author Comment (AC2) · 7 May 2020

Replies (R2) to reviewer # 2 (major (Maj#) comments) (anonymous, 16 Mar 2020) (line numbers are those of the initial submission)

Maj1. The presentation of the sequencing process employed is inadequate. The current text highlights that the sequences were run on a Illumina MiSeq, without providing additional details.

Maj1a : First, the study does not mention how the nucleic acids are extracted from the samples, checked for quality, stored, and shipped to the facility. These points must be

clarified.

R2-maj1a : The following sentences were added to clarify these issues.

From L138: "About 600 mg of sediments or soils, or up to 5 L of aquifer or runoff water samples filtered using 0.22 $\mu$m polycarbonate filters, were used per DNA extraction. Total DNAs were extracted from soils/sediments or filters using the FastDNA SPIN$^{®}$ Kit for Soil (MP Biomedicals, Carlsbad, France). For clay bead biofilms, microbial cells were detached by shaking at 2500 rpm for 2 min in 10 mL of 0.8 % NaCl. These suspensions were then filtered and their DNA content was extracted as indicated above. Blank samples were performed during these extractions for both the soils/sediments or filtered cells. DNAs were quantified using a nanodrop UV-Vis Spectrophotometer. Blank DNA extracts showed values below the detection limit. DNA extracts were visualized after electrophoresis at 6V/cm using a TBE buffer (89 mM Tri-borate, 89 mM boric acid, 2 mM EDTA, (pH 8.0)) through a 0.8% (w/v) agarose gel, and DNA staining with 0.4 mg.mL-1 ethidium bromide. A Gel Doc XR+ System (Bio-Rad, France) was used to observe the stained DNA, and confirm their relative quantities (between 20-120 ng/$\mu$l; median value around 40 ng/$\mu$L) and qualities. DNAs were kept at -80°C, and shipped on ice within 24h to the DNA sequencing services when appropriate.

Quantitative PCR assays were performed on the DNA extracts to estimate their relative content in 16S rRNA gene copies. These assays were performed on a Bio-Rad CFX96 realtime PCR instrument with Bio-Rad CFX Manager software, version 3.0 (Marnes-la-Coquette, France). The 16S rRNA gene primers 338F and 518R described by Park and Crowley (2006) were used, together with the Brilliant II SYBR green low ROX qPCR master mix for SYBR Green qPCR. Melting T° was 60°C. Linearized plasmid DNAs containing a 16S rRNA gene were used as standards, and obtained from Marti et al. (2017). Presence of inhibitors in the DNA extracts was checked by spiking known amount of plasmid harboring int2 (107 copies of plasmid per $\mu$L) in the PCR mix. Number of cycles needed to get a PCR signal was compared with wells where only plasmid DNA harboring int2 was added to the qPCR mix. When a high number of cycles was

needed to observe a signal, a 5- or 10-fold dilution of the DNA extract was done, and another round of tests was performed to confirm the absence of PCR inhibitions. Each assay was triplicated on distinct DNA extracts, and technical triplicates were performed. The 16S rRNA gene qPCR datasets are presented in Figure S1. These assays confirmed the high number of bacterial cells per compartment (Figure S1 and Table S2): (1) soils from the infiltration basin (IB) had a median content of $1.32 \times 10^{11}$ 16S rRNA gene copies per g dry weight; (2) sediments from the detention basin (DB) of $1.83 \times 10^{11}$ 16S rRNA gene copies per g dry weight, (3) the runoff waters (WS) had a median content of $4.75 \times 10^8$ 16S rRNA gene copies per mL, (4) the aquifer waters (AQ_wat) of $3.10 \times 10^6$ 16S rRNA gene copies per mL, and (5) the aquifer clay bead biofilms showed $1.35 \times 10^7$ 16S rRNA gene copies per cm2."

Maj1b : Second, the study must clarify within section 2.2 several key points with respect to the sequencing protocol: (1) a citation for the primers used to target the 16S rRNA gene, (2) the protocol followed by the laboratory must be unambiguously indicated or referenced (TruSeq, Nextera, etc.), (3) the target length of the sequences, and (4) whether the sequence reads were paired-end or single.

R2-maj1b : After the text added for comment R2-maj1a, the following sentences were added to clarify the Maj1b issues:

[revised manuscript text omitted]

Maj1c : Third, the presented study does not mention either positive mock community or negative comparison controls (and how those samples are incorporated into the analyses to remove contaminating sequences). The authors must present these controls.

R2-maj1c : As indicated above in replies "R2-maj1a" and "R2-maj1b", several blanks and lab controls were performed all over the investigations. Blanks were run during the DNA extractions, and did not yield detectable contaminant DNAs. Furthermore, the 16S rRNA gene qPCR datasets (Table S2) confirmed that high bacterial numbers were found among each compartment investigated in this study as indicated in reply "R2-maj1a". In fact, blanks were performed during the 799F - 1193R PCR amplifications of the V5-V6 16S rRNA gene regions, and DNA yields were found below the detection limit (<0,05 ng/$\mu$l). Any contaminant DNA would thus be highly diluted and not expected to have major incidence on this 16S rRNA gene-based meta-barcoding community

coalescence analysis. However, it is to be noted that the bacterial tpm community being expected to be in lower number per sample, blank samples for the tpm meta-barcoding sequencing scheme were sequenced. As indicated in "R2-maj1b", low number of tpm reads were obtained and their matching OTUs were listed in Table S3. These reads did not match tpm OTUs transferred from the above ground environments down into the aquifer.

To further clarify these issues, the following sentences were added:

From L294: It is to be noted that blank samples sequenced during the tpm meta-barcoding assay revealed 23 Pseudomonas OTUs coming from the DNA extraction kit or generated during the PCR product Illumina sequencing process (Table S3). Only OTU00573 was found in high number (867 reads) but this contaminant did not have an impact on the coalescence analysis because of its absence in the below ground datasets. Other contaminant OTUs did not represent more than 10 times the ones observed in the field samples for identical OTUs, a criterium used to distinguish significant contaminants (Lukasik et al., 2017; doi.org/10.1111/mec.14140). In fact, only seven OTUs found among the blanks matched OTUs recovered from the environmental samples, and only two of these could be related to well-defined species i. e. P. xanthomarina (17 reads among all blanks) and P. fragi (three reads among all blanks). These reads matched a single OTU over eleven allocated to P. xanthomarina in the environmental samples, and one OTU over 52 for P. fragi.

Maj2. The results of the sequencing campaign additionally requires a more comprehensive presentation. L193-194 presents the total sequencing reads, but must present the average and range of reads per sample. A supplemental table must be provided with the raw and processed sequencing counts for each sample.

R2-maj2 : These features are now indicated in Table S2, and cited in the text. From Line 193, the following sentence was added: "The analysis of the 16S rRNA V5-V6 gene libraries yielded 2,124,272 high-quality sequences distributed across 103 samples, as

described in Table S2.

Maj3. Additionally, to explore quantitatively the mixing ratios and why certain communities are providing more biomass, the actual concentration of the community within these compartments should be mentioned or addressed as to why these measurements were neglected.

R2-maj3 : The 16S rRNA gene qPCR datasets are now shown in Figure S1 and Table S2. They confirmed a lower number of bacterial cells among the aquifer than the runoff waters.

From L343, the following sentence was added: "...These results were confirmed by qPCR estimations of 16S rRNA gene copies per compartment. These values were much lower in the aquifer waters than the runoffs."

Maj4. The bioinformatic processing pipeline requires additional information. First, the approach presented divides the 16S rRNA amplicons into 97% OTUs. However, current best practices recommends utilizing the amplicon sequencing variants (ASV) approach (Knight et al., 2018).

maj4a : The authors should either update their approach to the ASV methodology or provide a concise defense as to why they selected the OTU approach.

maj4b : Second, a rarefaction analysis is presented to subsample the dataset at 20,624 sequences. This approach has been recently called into question for more directed comparisons (McMurdie and Holmes 2014). The authors should present a concise defense as to why rarefaction was employed. To bolster this defense, Figure S1 should display the rarefaction curve for the raw data, not the previously subsampled 20,624 dataset (this comment connects with Maj2 in the need to present additional information).

R2-maj4a and 4b : Figure S1 was replaced by Figure S2 which is now showing both the OTU rarefaction curves before and after having performed a sub-sampling

at 20,624 reads per sample. OTUs were defined at a 97% identity cut-off to collapse reads into groups that reduce the incidence of sequencing errors on the dataset as suggested by several authors including Eren et al. (2013; PLOS ONE 8, doi : 10.1371/journal.pone.0066643), and Johnson et al. (2019; Nat. Commun. 10:5029, doi: 10.1038/s41467-019-13036-1).

It is to be noted that the original paper by Knights et al. (2011) describing the development of the SourceTracker made use of OTU contingency tables built with a 97% identity cut-off. This was also the case of the paper describing a "reliability" test for the source tracker inferences (Henry et al., 2016; https://doi.org/10.1016/j.watres.2016.02.029). Looking at recently published papers on the SourceTracker, one can find that most research groups have maintained a use of OTU-based contingency tables e. g. O'Dea et al. (2019, https://doi.org/10.1016/j.watres.2019.114967); Han et al. (2020, https://doi.org/10.1016/j.watres.2020.115469), Chen et al. (2019, https://doi.org/10.1038/s41598-019-42548-5), Bi et al. (2019, doi:10.1111/1462-2920.14614), and so on. Still, we confirm that a few papers have used the ASV approach to build their contingency tables for the SourceTracker and for other purposes e. g. Karstens et al. 2019, https:// doi.org/10.1128/mSystems.00290-19, and Caruso et al., 2019; https:// doi.org/10.1128/mSystems.00163-18. We recognize that the ASV approach is reliable to identify conserved ASV among datasets showing variable number of reads. However, the ASV approach also has its weaknesses. For our actual application of the SourceTracker, and according to other papers, the OTU-based contingency table was thus kept for our downstream analyses. Nevertheless, we've now cited articles on ASV in order to make sure that future readers of this paper will be aware of this approach, and might consider using it for the SourceTracker analyses.

The sub-sampling performed at 20,624 reads allowed to reduce the incidence of the variable number of reads obtained per sample. An uneven sequencing depth (ranging from 6,062 to 181,207 reads per sample) was recorded, and found to be related to

technical DNA sequencing problems. In fact, the qPCR datasets on 16S rRNA gene copies supported this conclusion. No correlation was observed between the 16S rRNA gene copy numbers (biomass) and the number of reads obtained per sample (see Table S2). In this context, we've decided to sub-sample our dataset to compensate for these discrepancies. In our opinion, sub-sampling datasets remain a good standardization technique to mitigate sample library size artifacts, especially for very unequal library sizes between groups. In accordance with this, our sub-sampled dataset (20,624 reads per sample) led to a very good separation of samples according to their origin (i.e. WS, DB, IB, AQ_wat and AQ_bio) (see Fig. 3).

From 155, the following sentences were added to clarify these issues: Variability in the number of cleaned reads per sample was observed but not correlated with variations in the number of 16S rRNA gene sequences (Table S2). These variations were thus considered to be due to the DNA sequencing process. Therefore, a sub-sampled dataset (20,624 reads per sample; with exclusion of samples with total reads below this threshold) was used to mitigate the artifact of sample library sizes. Operational Taxonomic Units (OTUs) were defined using a 97% identity cut-off as recommended by several authors in order to collapse sequences into groups that reduce the incidence of sequence errors on the datasets (e. g., Eren et al. 2013; and Johnson et al. 2019). It is to be noted that amplicon sequence variants (ASV) could also be used to build contingency tables (e. g., Callahan et al. 2016; Karstens et al. 2019). However, exact sequence variants can generate uncertainties when using 16S rRNA gene sequences because of variations among species and strains due to the presence of multiple copies per genome (Johnson et al. 2019). Figure S2 shows the OTU rarefaction curves for the full and the sub-sampled datasets. This sub-sampled dataset was used for all downstream analyses except those of the SourceTracker Bayesian approach.

Maj5. In the SourceTracker default code, the rarefied sample is then rarefied further to 1000. This procedure should be repeated to draw those 1000 reads from the full dataset, not the previously rarefied data.

[Figure]

R2-maj5 : We agree with this comment. Analyses were thus re-run using the cleaned but not re-sampled 16S rRNA gene reads, and the matching OTU contingency table (the one used to build Figure S2a). We then used the default SourceTracker code, including a sub-sampling of 1,000 reads as recommended by Henry et al. (2016). This analysis was run 3 times, and the coefficient of variation (i.e. Relative Standard Deviation) was used as a gauge to evaluate confidence on the computed values as suggested by Henry et al. (2016) and McCarthy et al. (2017). Table 1 was modified according to these computings.

Maj6. L319-337 presents a great overview of the study that is more appropriate for the abstract rather than the discussion. This section should be removed in its entirety.

R2-maj6 : This paragraph was deleted but a few sentences kept to facilitate the understanding of the discussion

Maj7. Throughout the text, the presence of a specific 16S rRNA transcript often is utilized to state the presence of a specific function within the community, notably within the abstract (e.g., L25, L27). Whereas the 16S taxonomical assignment is a good indicator that a specific function is likely encoded on the metagenome of the community, the linkage is not directly shown through the 16S survey and must be caveated by "likely", "putative", or "predicted to be". This is recognized more consistently within the discussion of the results, but must be maintained throughout the text to recognize that the assignment provided by FAPROTAX is a hypothesis.

R2-maj7 : Ok, this was clarified over the text.

Maj8. The authors commendably provided the raw data as publicly available datasets through EBI. Additionally, the authors should provide all code utilized to process these data as a part of the supplemental materials to allow future readers to reconstruct the presented results.

R2-maj8 : From L149, the following sentences were added so that future readers

can reproduce the results generated in this work : All paired-end MiSeq reads were processed using Mothur 1.40.4 by following a standard operation protocol (SOP) for MiSeq-based microbial community analysis (Schloss et al., 2009; Kozich et al.(2013), so-called MiSeq SOP available at http://www.mothur.org/wiki/MiSeq_SOP. Due to the large number of sequences to be processed, the cluster.split command was used to assign sequences to OTUs.

Maj9. The authors are encouraged to focus on improving the English language and grammar associated with the presented article. A non-exhaustive list of suggested grammar improvements is provided in the final section of this review, but additional editing services are recommended to enhance the clarity and accuracy of the text.

R2-maj9 : we did a complete grammar review and rewrote some sentences to clarify certain formulations.

Minor Min1. The bulk physical and chemical properties of the sampling sites should be presented or directly cited such as pH, temperature, electroconductivity etc.

reply : fixed; the most significant chemical datasets are now indicated in the paper from L365; and a selection of papers was cited so that readers can complete their knowledge of the investigated sites through analysis of these papers which present pH, electrical conductivity, soil properties, and many other datasets. See replies to reviewer 1 for this issue.

Additionally, please replace "for which physico-chemical and biological monitorings have been implemented" with "that records both physico-chemical and biological properties."

reply : fixed accordingly

Min2. L34 – Please clarify what is meant by "DNA imprints allocated"

reply : was changed for "Some tpm sequence types of . . ."

Min3. L70-75 – Please provide citations in support of these claims.

reply : fixed

Min4. L78-L79 – Replace "The tested hypotheses were that" with "Two hypotheses were tested:".

reply : fixed

Because these statements are presenting the underlying hypotheses (supported or rejected), all qualifiers for the verbs must be removed. Therefore, remove L78 "should" and L79 "could also". L79 – Replace "but" with ", and".

reply : fixed accordingly

Similarly with L88-90, please replace "was likely to be" with "will be"

reply : fixed accordingly

Min5. L291-307 – The long list of species mapped to the Pseudomonas genera is difficult to interpret in the currently presented form. Please condense this section for readability.

reply : we've tried to simplify this text but citing all these species is important for specialists; several of these species had never been described in these environmental contexts or in Europe

Min6. Throughout the text, ensure that a comma appears after Latin abbreviations such as i.e., and e.g.,

reply : fixed accordingly

Min7. Figure 1, please italicize the names of the phyla.

reply : fixed accordingly

Grammar / reply: all grammar issues raised by this reviewer were considered and fixed.

Please also note the supplement to this comment:
https://www.hydrol-earth-syst-sci-discuss.net/hess-2020-39/hess-2020-39-AC2-supplement.pdf

———————————————————

[Figure]

Colin et al. revised Table 1

**Table 1.** Coalescence of surface and aquifer bacterial communities inferred by the SourceTracker Bayesian approach and the 16S rRNA gene meta-barcoding dataset*

| samples | | WS | | DB | | IB | | AQ_wat_up | | unknown | |
|---|---|---|---|---|---|---|---|---|---|---|---|
| | | mean | rsd | mean | rsd | mean | rsd | mean | rsd | mean | rsd |
| 1 - waters | AQ_wat_dw1 | 0.3% | 33.3 | 0.3% | 43.3 | 7.5% | 42.6 | 19.7% | 30.6 | 72.3% | 4.8 |
| | AQ_wat_dw2 | 10.2% | 50.6 | 17.6% | 10.3 | 9.7% | 18.8 | 25.7% | 15.4 | 36.9% | 6.7 |
| | AQ_wat_dw3 | 5.0% | 9.0 | 5.0% | 29.1 | 3.8% | 32.0 | 70.7% | 1.9 | 15.5% | 2.3 |
| | AQ_bio_dw1 | 8.6% | 23.5 | 25.0% | 19.1 | 3.9% | 74.1 | 56.7% | 6.9 | 5.8% | 7.9 |
| | AQ_bio_dw2 | 13.6% | 28.0 | 28.4% | 14.1 | 2.9% | 46.0 | 48.2% | 6.52 | 6.8% | 11.8 |
| 1 - biofilms | AQ_bio_dw3 | 3.4% | 17.1 | 13.9% | 18.4 | 5.5% | 39.3 | 72.1% | 1.85 | 5.2% | 29.8 |
| 2 - biofilms | AQ_bio_up1 | 32.2% | 14.5 | | | | | 61.3% | 9.5 | 6.8% | 23.7 |
| | AQ_bio_up2 | 56.6% | 12.6 | | | | | 36.4% | 18.3 | 7.0% | 15.7 |
| | AQ_bio_up3 | 44.0% | 6.6 | | | | | 48.1% | 8.1 | 7.8% | 10.8 |

* Two analyses are shown from sub-sampled datasets set at 1000 reads: (1) reads from WS, DB, IB, and aquifer waters from upstream the SIS were considered as the sources of taxa for the aquifer samples downstream the SIS; (2) reads from WS and the aquifer waters upstream the SIS were considered as the sources of taxa for the aquifer biofilms recovered upstream the SIS. SourceTracker was run 3 times using the 16S rRNA gene OTU contingency table and the default parameters. Relative contributions of the sources were averaged. Relative standard deviations (%RSD) are indicated, and used as confidence values. RSD > 100% indicates low confidence on the estimated value. WS: Watershed zone of waters; DB: Detention basin sediments; IB: Infiltration basin sediments. Sequences that could not be attributed to one of the tested sources were grouped under the term unknown.

**Fig. 1.** revised Table 1

Figure S1. Boxplot representation of the 16S rRNA gene copy numbers measured by quantitative PCR per DNA extracts of runoff waters (WS), sediments from the detention basin (DB), soils from the infiltration basin (IB), aquifer waters (AQ_waters) or aquifer clay beads biofilms (AQ_bio). Values were expressed per g of dry weight soil or sediment, or per mL, or per surface for the clay bead biofilms.

**Fig. 2.** new Suppl. Fig S1

[Figure]

Figure S2. Rarefaction curves showing the relation between the number of V5-V6 16S rRNA (*rrs*) gene reads analyzed and OTU numbers per compartment of the Mi-plaine watershed of Chassieu (France). (a) without sub-sampling and (b) with a sub-sampling performed at 20,624 reads per sample.

**Fig. 3.** new Suppl. Fig S2

**Fig. 4.** new Suppl. Table S2

Table S3. Number of *tpm* reads among blank samples run during the tpm meta-barcoding procedure, and their taxonomic allocation and relatedness to OTUs recovered from the environmental samples. *: restricted to above ground samples; **: not considered in the coalescence analysis, see Table S8.

| blank sample OTU | total number of reads | identical OTU sequence among the environmental samples | maximum % identity with environmental *tpm* sequences | genus | species | blank 1 (soil) | blank 2 (soil) | blank 3 (water) | blank 4 (water) | blank 5 (water) |
|---|---|---|---|---|---|---|---|---|---|---|
| Otu01 | 867 | Otu00573* | 100 | Pseudomonas | unclassified | 0 | 0 | 0 | 867 | 0 |
| Otu02 | 118 | | 99 | Pseudomonas | fluorescens | 0 | 0 | 0 | 118 | 0 |
| Otu03 | 21 | | 99 | Pseudomonas | fluorescens | 21 | 0 | 0 | 0 | 0 |
| Otu04 | 17 | | no match | unclassified | unclassified | 1 | 0 | 15 | 0 | 0 |
| Otu05 | 17 | Otu00151* | 100 | Pseudomonas | xanthomarina | 0 | 0 | 8 | 9 | 0 |
| Otu06 | 13 | | no match | unclassified | unclassified | 1 | 0 | 12 | 0 | 0 |
| Otu07 | 10 | | 99 | Pseudomonas | unclassified | 0 | 0 | 0 | 10 | 0 |
| Otu08 | 7 | | no match | unclassified | unclassified | 0 | 1 | 6 | 0 | 0 |
| Otu09 | 7 | | 99 | Pseudomonas | unclassified | 0 | 0 | 0 | 7 | 0 |
| Otu10 | 6 | | no match | unclassified | unclassified | 1 | 0 | 5 | 0 | 0 |
| Otu11 | 5 | | no match | unclassified | unclassified | 0 | 0 | 5 | 0 | 0 |
| Otu12 | 4 | Otu01054** | 100 | unclassified | unclassified | 0 | 0 | 0 | 3 | 0 |
| Otu13 | 3 | | 99 | Pseudomonas | unclassified | 0 | 0 | 0 | 3 | 0 |
| Otu14 | 3 | Otu00069 | 100 | Pseudomonas | fragi | 0 | 0 | 3 | 0 | 0 |
| Otu15 | 3 | Otu00002* | 100 | Pseudomonas | unclassified | 0 | 0 | 2 | 1 | 0 |
| Otu16 | 2 | | 99 | Pseudomonas | unclassified | 0 | 0 | 0 | 2 | 0 |
| Otu17 | 2 | | no match | unclassified | unclassified | 0 | 0 | 0 | 1 | 0 |
| Otu18 | 2 | | 98 | unclassified | unclassified | 0 | 0 | 2 | 0 | 0 |
| Otu19 | 2 | Otu00519** | 100 | unclassified | unclassified | 0 | 0 | 0 | 1 | 1 |
| Otu20 | 2 | | 99 | Pseudomonas | unclassified | 0 | 0 | 0 | 2 | 0 |
| Otu21 | 2 | | 99 | Pseudomonas | unclassified | 0 | 0 | 0 | 2 | 0 |
| Otu22 | 2 | Otu00556** | 100 | unclassified | unclassified | 0 | 2 | 0 | 0 | 0 |
| Otu23 | 2 | | 99 | Pseudomonas | unclassified | 0 | 0 | 0 | 2 | 0 |

**Fig. 5.** new Suppl. Table S3

**Supplement:**

**Replies (R2) to reviewer # 2 (major (Maj#) comments) (anonymous, 16 Mar 2020)**
(line numbers are those of the initial submission)

Maj1. The presentation of the sequencing process employed is inadequate. The current text highlights that the sequences were run on a Illumina MiSeq, without providing additional details.

> Maj1a : First, the study does not mention how the nucleic acids are extracted from the samples, checked for quality, stored, and shipped to the facility. These points must be clarified.

> **R2-maj1a** :  The following sentences were added to clarify these issues.

> From L138: "About 600 mg of sediments or soils, or up to 5 L of aquifer or runoff water samples filtered using 0.22 μm polycarbonate filters, were used per DNA extraction. Total DNAs were extracted from soils/sediments or filters using the FastDNA SPIN® Kit for Soil (MP Biomedicals, Carlsbad, France). For clay bead biofilms, microbial cells were detached by shaking at 2500 rpm for 2 min in 10 mL of 0.8 % NaCl. These suspensions were then filtered and their DNA content was extracted as indicated above. Blank samples were performed during these extractions for both the soils/sediments or filtered cells. DNAs were quantified using a nanodrop UV-Vis Spectrophotometer. Blank DNA extracts showed values below the detection limit. DNA extracts were visualized after electrophoresis at 6V/cm using a TBE buffer (89 mM Tri-borate, 89 mM boric acid, 2 mM EDTA, (pH 8.0)) through a 0.8% (w/v) agarose gel, and DNA staining with 0.4 mg.mL-1 ethidium bromide. A Gel Doc XR+ System (Bio-Rad, France) was used to observe the stained DNA, and confirm their relative quantities (between 20-120 ng/μl; median value around 40 ng/μL) and qualities. DNAs were kept at -80°C, and shipped on ice within 24h to the DNA sequencing services when appropriate.

> Quantitative PCR assays were performed on the DNA extracts to estimate their relative content in 16S rRNA gene copies. These assays were performed on a Bio-Rad CFX96 realtime PCR instrument with Bio-Rad CFX Manager software, version 3.0 (Marnes-la-Coquette, France). The 16S rRNA gene primers 338F and 518R described by Park and Crowley (2006) were used, together with the Brilliant II SYBR green low ROX qPCR master mix for SYBR Green qPCR. Melting T° was 60°C. Linearized plasmid DNAs containing a 16S rRNA gene were used as standards, and obtained from Marti et al. (2017). Presence of inhibitors in the DNA extracts was checked by spiking known amount of plasmid harboring *int2* ($10^7$ copies of plasmid per μL) in the PCR mix. Number of cycles needed to get a PCR signal was compared with wells where only plasmid DNA harboring *int2* was added to the qPCR mix. When a high number of cycles was needed to observe a signal, a 5- or 10-fold dilution of the DNA extract was done, and another round of tests was performed to confirm the absence of PCR inhibitions. Each assay was triplicated on distinct DNA extracts, and technical triplicates were performed. The 16S rRNA gene qPCR datasets are presented in Figure S1. These assays confirmed the high number of bacterial cells per compartment (Figure S1 and Table S2): (1) soils from the infiltration basin (IB) had a median content of $1.32 \times 10^{11}$ 16S rRNA gene copies per g dry weight; (2) sediments from the detention basin (DB) of $1.83 \times 10^{11}$ 16S rRNA gene copies per g dry weight, (3) the runoff waters (WS) had a median content of $4.75 \times 10^8$ 16S rRNA gene copies per mL, (4) the aquifer waters (AQ_wat) of $3.10 \times 10^6$ 16S rRNA gene copies per mL, and (5) the aquifer clay bead biofilms showed $1.35 \times 10^7$ 16S rRNA gene copies per cm$^2$."

Maj1b : Second, the study must clarify within section 2.2 several key points with respect to the sequencing protocol: (1) a citation for the primers used to target the 16S rRNA gene, (2) the protocol followed by the laboratory must be unambiguously indicated or referenced (TruSeq, Nextera, etc.), (3) the target length of the sequences, and (4) whether the sequence reads were paired-end or single.

> **R2-maj1b** : After the text added for comment **R2-maj1a**, the following sentences were added to clarify the Maj1b issues:

[revised manuscript text omitted]

Maj1c : Third, the presented study does not mention either positive mock community or negative comparison controls (and how those samples are incorporated into the analyses to remove contaminating sequences). The authors must present these controls.

**R2-maj1c** : As indicated above in replies "R2-maj1a" and R2-maj1b, several blanks and lab controls were performed all over the investigations. Blanks were run during the DNA extractions, and did not yield detectable contaminant DNAs. Furthermore, the 16S rRNA gene qPCR datasets (Table S2) confirmed that high bacterial numbers were found among each compartment investigated in this study as indicated in reply "R2-maj1a". In fact, blanks were performed during the 799F - 1193R PCR amplifications of the V5-V6 16S rRNA gene regions, and DNA yields were found below the detection limit (<0,05 ng/µl). Any contaminant DNA would thus be highly diluted and not expected to have major incidence on this 16S rRNA gene-based meta-barcoding community coalescence analysis. However, it is to be noted that the bacterial *tpm* community being expected to be in lower number per sample, blank samples for the *tpm* meta-barcoding sequencing scheme were sequenced. As indicated in "R2-maj1b", low number of *tpm* reads were obtained and their matching OTUs were listed in Table S3. These reads did not match *tpm* OTUs transferred from the above ground environments down into the aquifer.

To further clarify these issues, the following sentences were added:

From L294:

It is to be noted that blank samples sequenced during the *tpm* meta-barcoding assay revealed 23 *Pseudomonas* OTUs coming from the DNA extraction kit or generated during the PCR product Illumina sequencing process (Table S3). Only OTU00573 was found in high number (867 reads) but this

contaminant did not have an impact on the coalescence analysis because of its absence in the below ground datasets. Other contaminant OTUs did not represent more than 10 times the ones observed in the field samples for identical OTUs, a criterium used to distinguish significant contaminants (Lukasik et al., 2017; doi.org/10.1111/mec.14140). In fact, only seven OTUs found among the blanks matched OTUs recovered from the environmental samples, and only two of these could be related to well-defined species i. e. *P. xanthomarina* (17 reads among all blanks) and *P. fragi* (three reads among all blanks). These reads matched a single OTU over eleven allocated to *P. xanthomarina* in the environmental samples, and one OTU over 52 for *P. fragi*.

Maj2. The results of the sequencing campaign additionally requires a more comprehensive presentation. L193-194 presents the total sequencing reads, but must present the average and range of reads per sample. A supplemental table must be provided with the raw and processed sequencing counts for each sample.

**R2-maj2** : These features are now indicated in Table S2, and cited in the text.

From Line 193, the following sentence was added:

"The analysis of the 16S rRNA V5-V6 gene libraries yielded 2,124,272 high-quality sequences distributed across 103 samples, as described in Table S2.

Maj3. Additionally, to explore quantitatively the mixing ratios and why certain communities are providing more biomass, the actual concentration of the community within these compartments should be mentioned or addressed as to why these measurements were neglected.

**R2-maj3** : The 16S rRNA gene qPCR datasets are now shown in Figure S1 and Table S2. They confirmed a lower number of bacterial cells among the aquifer than the runoff waters.

From L343, the following sentence was added:

"…These results were confirmed by qPCR estimations of 16S rRNA gene copies per compartment. These values were much lower in the aquifer waters than the runoffs."

Maj4. The bioinformatic processing pipeline requires additional information. First, the approach presented divides the 16S rRNA amplicons into 97% OTUs. However, current best practices recommends utilizing the amplicon sequencing variants (ASV) approach (Knight et al., 2018).

maj4a : The authors should either update their approach to the ASV methodology or provide a concise defense as to why they selected the OTU approach.

maj4b : Second, a rarefaction analysis is presented to subsample the dataset at 20,624 sequences. This approach has been recently called into question for more directed comparisons (McMurdie and Holmes 2014). The authors should present a concise defense as to why rarefaction was employed. To bolster this defense, Figure S1 should display the rarefaction curve for the raw data, not the previously subsampled 20,624 dataset (this comment connects with Maj2 in the need to present additional information).

**R2-maj4a and 4b** :

Figure S1 was replaced by Figure S2 which is now showing both the OTU rarefaction curves before and after having performed a sub-sampling at 20,624 reads per sample. OTUs were defined at a 97% identity cut-off to collapse reads into groups that reduce the incidence of sequencing errors on the dataset as suggested by several authors including Eren et al. (2013; PLOS ONE 8, doi : 10.1371/journal.pone.0066643), and Johnson et al. (2019; Nat. Commun. 10:5029, doi: 10.1038/s41467-019-13036-1).

It is to be noted that the original paper by Knights et al. (2011) describing the development of the SourceTracker made use of OTU contingency tables built with a 97% identity cut-off. This was also the case of the paper describing a "reliability" test for the source tracker inferences (Henry et al., 2016; https://doi.org/10.1016/j.watres.2016.02.029). Looking at recently published papers on the SourceTracker, one can find that most research groups have maintained a use of OTU-based contingency tables e. g. O'Dea et al. (2019, https://doi.org/10.1016/j.watres.2019.114967); Han et al. (2020, https://doi.org/10.1016/j.watres.2020.115469), Chen et al. (2019, https://doi.org/10.1038/s41598-019-

42548-5), Bi et al. (2019, doi:10.1111/1462-2920.14614), and so on. Still, we confirm that a few papers have used the ASV approach to build their contingency tables for the SourceTracker and for other purposes e. g. Karstens et al. 2019, https:// doi.org/10.1128/mSystems.00290-19, and Caruso et al., 2019; https:// doi.org/10.1128/mSystems.00163-18. We recognize that the ASV approach is reliable to identify conserved ASV among datasets showing variable number of reads. However, the ASV approach also has its weaknesses. For our actual application of the SourceTracker, and according to other papers, the OTU-based contingency table was thus kept for our downstream analyses. Nevertheless, we've now cited articles on ASV in order to make sure that future readers of this paper will be aware of this approach, and might consider using it for the SourceTracker analyses.

The sub-sampling performed at 20,624 reads allowed to reduce the incidence of the variable number of reads obtained per sample. An uneven sequencing depth (ranging from 6,062 to 181,207 reads per sample) was recorded, and found to be related to technical DNA sequencing problems. In fact, the qPCR datasets on 16S rRNA gene copies supported this conclusion. No correlation was observed between the 16S rRNA gene copy numbers (biomass) and the number of reads obtained per sample (see Table S2). In this context, we've decided to sub-sample our dataset to compensate for these discrepancies. In our opinion, sub-sampling datasets remain a good standardization technique to mitigate sample library size artifacts, especially for very unequal library sizes between groups. In accordance with this, our sub-sampled dataset (20,624 reads per sample) led to a very good separation of samples according to their origin (i.e. WS, DB, IB, AQ_wat and AQ_bio) (see Fig. 3).

From 155, the following sentences were added to clarify these issues:

Variability in the number of cleaned reads per sample was observed but not correlated with variations in the number of 16S rRNA gene sequences (Table S2). These variations were thus considered to be due to the DNA sequencing process. Therefore, a sub-sampled dataset (20,624 reads per sample; with exclusion of samples with total reads below this threshold) was used to mitigate the artifact of sample library sizes. Operational Taxonomic Units (OTUs) were defined using a 97% identity cut-off as recommended by several authors in order to collapse sequences into groups that reduce the incidence of sequence errors on the datasets (e. g., Eren et al. 2013; and Johnson et al. 2019). It is to be noted that amplicon sequence variants (ASV) could also be used to build contingency tables (e. g., Callahan et al. 2016; Karstens et al. 2019). However, exact sequence variants can generate uncertainties when using 16S rRNA gene sequences because of variations among species and strains due to the presence of multiple copies per genome (Johnson et al. 2019). Figure S2 shows the OTU rarefaction curves for the full and the sub-sampled datasets. This sub-sampled dataset was used for all downstream analyses except those of the SourceTracker Bayesian approach.

Maj5. In the SourceTracker default code, the rarefied sample is then rarefied further to 1000. This procedure should be repeated to draw those 1000 reads from the full dataset, not the previously rarefied data.

**R2-maj5** :

We agree with this comment. Analyses were thus re-run using the cleaned but not re-sampled 16S rRNA gene reads, and the matching OTU contingency table (the one used to build Figure S2a). We then used the default SourceTracker code, including a sub-sampling of 1,000 reads as recommended by Henry et al. (2016). This analysis was run 3 times, and the coefficient of variation (i.e. Relative Standard Deviation) was used as a gauge to evaluate confidence on the computed values as suggested by Henry et al. (2016) and McCarthy et al. (2017). Table 1 was modified according to these computings.

Maj6. L319-337 presents a great overview of the study that is more appropriate for the abstract rather than the discussion. This section should be removed in its entirety.

**R2-maj6** : This paragraph was deleted but a few sentences kept to facilitate the understanding of the discussion

Maj7. Throughout the text, the presence of a specific 16S rRNA transcript often is utilized to state the presence of a specific function within the community, notably within the abstract (e.g., L25, L27). Whereas the 16S

taxonomical assignment is a good indicator that a specific function is likely encoded on the metagenome of the community, the linkage is not directly shown through the 16S survey and must be caveated by "likely", "putative", or "predicted to be". This is recognized more consistently within the discussion of the results, but must be maintained throughout the text to recognize that the assignment provided by FAPROTAX is a hypothesis.

**R2-maj7** : Ok, this was clarified over the text.

Maj8. The authors commendably provided the raw data as publicly available datasets through EBI. Additionally, the authors should provide all code utilized to process these data as a part of the supplemental materials to allow future readers to reconstruct the presented results.

**R2-maj8** :

From L149, the following sentences were added so that future readers can reproduce the results generated in this work :

All paired-end MiSeq reads were processed using Mothur 1.40.4 by following a standard operation protocol (SOP) for MiSeq-based microbial community analysis (Schloss et al., 2009; Kozich *et al.*(2013), so-called MiSeq SOP available at http://www.mothur.org/wiki/MiSeq_SOP. Due to the large number of sequences to be processed, the cluster.split command was used to assign sequences to OTUs.

Maj9. The authors are encouraged to focus on improving the English language and grammar associated with the presented article. A non-exhaustive list of suggested grammar improvements is provided in the final section of this review, but additional editing services are recommended to enhance the clarity and accuracy of the text.

**R2-maj9** : we did a complete grammar review and rewrote some sentences to clarify certain formulations.

Minor

Min1. The bulk physical and chemical properties of the sampling sites should be presented or directly cited such as pH, temperature, electroconductivity etc.

**reply** : fixed; the most significant chemical datasets are now indicated in the paper from L365; and a selection of papers was cited so that readers can complete their knowledge of the investigated sites through analysis of these papers which present pH, electrical conductivity, soil properties, and many other datasets. See replies to reviewer 1 for this issue.

Additionally,

please replace "for which physico-chemical and biological monitorings have been implemented" with "that records both physico-chemical and biological properties."

**reply** : fixed accordingly

Min2. L34 – Please clarify what is meant by "DNA imprints allocated"

**reply** : was changed for "Some *tpm* sequence types of …"

Min3. L70-75 – Please provide citations in support of these claims.

**reply** : fixed

Min4. L78-L79 – Replace "The tested hypotheses were that" with "Two hypotheses were tested:".

**reply** : fixed

Because these statements are presenting the underlying hypotheses

(supported or rejected), all qualifiers for the verbs must be removed.

Therefore, remove

L78 "should" and L79 "could also". L79 – Replace "but" with ", and".

    **reply** : fixed accordingly

Similarly with L88-90, please replace "was likely to be" with "will be"

    **reply** : fixed accordingly

Min5. L291-307 – The long list of species mapped to the Pseudomonas genera is

difficult to interpret in the currently presented form. Please condense this section for

readability.

    **reply** : we've tried to simplify this text but citing all these species is important for specialists; several of these species had never been described in these environmental contexts or in Europe

Min6. Throughout the text, ensure that a comma appears after Latin abbreviations

such as i.e., and e.g.,

    **reply** : fixed accordingly

Min7. Figure 1, please italicize the names of the phyla.

    **reply** : fixed accordingly

Grammar / **reply:** all the points below were considered and fixed.

L23 – Please add "basins" after "detention". Currently, this sentence presents a broken

list of items.

L24 – Please replace "made up" with "comprised"

L27 – Please add a comma before "but a higher"

L28 – Please replace "a tracking" with "the tracking"

L29 – Please replace "including the" with "including", remove "among these communities", and replace "the Pseudomonas" with "Pseudomonas"

L31 – Please replace "respectively, of" with "respectively, in"

L32 – Please remove the comma before ", and waters" and add "to be" after "found"

L34 – Please add a comma before "but only"

L36 – Please add "in" after "than"

L48 – Please replace "during" with "over"

L53 – Please replace "towards" with "capturing" and remove the comma after "metals)"

L59-60 – Please add "both the" before "water transit", replace ", but also the biology"

with " and biological properties", replace "cover and root" with "cover, root", replace

"worms" with "worm population", and add "composition" after microbiota.

L76 – Please replace "Here, the" with "This", replace "explored" with "explores", and

remove the commas around ", with a thick vadose zone (> 10 m),"

L83 – Please replace "It" with "The site"

L84-85 – Please replace "It has an average vadose thickness of" with "The average

vadose thickness of the site is"

L86-88 – Please remove "large", replace "built" with "recorded", remove ", in order",
add "the" before "bacterial community", add "the" before "top", replace "among" with
"into", and add "the" before "biofilm".

L92 – Please replace "among" with "within" and replace "while" with ", whereas"

L99 – Please remove "To go deeper into these inferences," and replace "were built"
with "were then assembled"

L101 – Please remove the comma after "level," and replace ", and allowed gaining
further insights on" with "to explore with a higher resolution"

L103 – Please replace "with" with "within"

L108 – Please replace "about" with "approximately"

L110 – Please add "a" before "part"

L112 – Please remove the comma before ", built"

L113-114 – Please replace "development of a plant cover" with "plant cover development

L116 – Please replace "deeply" with "previously" and check the format requirements
for citations.

L118 – Please replace "were considered for this" with "are investigated within"

L120-122 – Please replace "have been" with "were", add an "a" before "50", and remove
"of the DB"

L122 – Please remove the comma before the "and"

L124 – Please replace "had been" with "were"

L126 – Please remove the comma before "at a"

L127 – Please replace "have been" with "were"

L130 – Please replace "had been" with "was"

L132 – Please add "subsequently" before "discarded"

L133 – Please replace "using clay" with "from clay"

L134 – Please replace "the same piezometers as those for the aquifer water samplings"
with "the piezometers described above" and delete the subsequent sentence whereas
including the citation just after (n = 6 samples).

L137 – Please revise this title to be more informative, such as "Generation and sequencing of the DNA amplicons"

L139 – Please replace "with Illumina MiSeq technology" with "on an Illumina MiSeq"

L165 – Please capitalize "BLAST"

L166 – Please remove "in order"

L168 – Please replace "carried out" with "performed"

L178 – Please remove "down"

L179 – Please remove ", in order"

L185 – Pease remove the comma before ", with"

L199 – Please replace "superior" with "greater than"

L205 – Please replace "of detention" with "withdrawn from the detention"

L207 – Please add a "the" before "clay" and replace "for" with "over"

L212 – Please replace of "to be made of" with "to contain"

L213 – Please replace "a same" with "the same"

L214 – Please replace "while" with "whereas the"

L216 – Please replace "found" with "to be found"

L226 – Please replace "while" with "whereas"

L233 – Please replace "much to" with "substantially to the" and remove "Content of"

L236 – Please replace "even though" with "although"

L238 – Please remove "In order"

L244 – Please add "and" before "Nitratireductor."

L245 – Please replace "while" with ", and"

L260 – Please replace "the SIS" with "of the SIS"

L261 – Please add "and" before "Flavobacterium"

L262 – Please replace "while" with "whereas"

L264 – Please add "and" before "Meganema"

L265 – Please replace "found: with "found to be"

L276 – Please replace "It is to be noted that" with "Notably,"

L277 – Please replace "part" with "representative"

L280 – Please replace "deeper" with "further"

L287 – Please remove "the"; additionally, because the data is already given as Table S6, I recommend removing the exact percentages from this paragraph.

L295 – Please add a comma before "but"

L296 – Please add a comma before "but"

L316 – Please replace "while" with "when"

L345 – Please remove the comma before ", and"

L347 – Please replace "that can also enhance" with "enhances"

L348 – Please remove "Nevertheless" and replace "has induced" with "induces"

L350 – Please replace "that" with "than"

L352 – Please replace "the SIS" with "of the SIS"

L353 – Please replace "the SIS" with "of the SIS"

L360 – Please replace "the SIS" with "of the SIS"

L385 – Please add a comma before ", and significant"

L389 – Please replace "while" with "whereas"

L391 – Please replace "to likely" with "likely"

L396 – Please replace "the SIS" with "of the SIS"

L398 – Please remove the comma before ", and was"

L400 – Please add a comma before "but a few"

L404 – Please replace "in order to go deeper into" with "to explore further"

L414 – Please remove the comma before ", and can"

L418 – Please replace "It would thus be part of the r-strategists that could get opportunistically established" with "Therefore, an r-strategist would likely establish opportunistically"

L422 – Please remove "However,"

L425 – Please add "to be" after "shown"

L429 – Please remove the comma before ", and yield"

L431 - Please add "to be" after "found"

L432 – Please replace "at degrading" with "to degrade"

L436 – Please use an alternative term to "germcatchers"

L437 – Please remove "down"

L439-440 – Please improve the wording of the sentence beginning with "Free-living"

L443 – Please remove "to these"

L687 – Please replace "runoffs" with "runoff"

L688-689 – Please remove the commas after "(A)," and "(B),"

L700 – Remove "down the aquifer"

Table 1. Capitalize "Downstream SIS"; in the caption, replace "in (A),", "in (B)," and "in (C)," with "(A)", "(B)", and "(C)", respectively. Remove all occurrences of "is indicated"

Table 1. Coalescence of surface and aquifer bacterial communities inferred by the SourceTracker Bayesian approach and the 16S rRNA gene meta-barcoding dataset*

| | samples | WS | | DB | | IB | | AQ_wat_up | | unknown | |
|---|---|---|---|---|---|---|---|---|---|---|---|
| | | mean | rsd | mean | rsd | mean | rsd | mean | rsd | mean | rsd |
| 1 - waters | AQ_wat_dw1 | 0.3% | 33.3 | 0.3% | 43.3 | 7.5% | 42.6 | 19.7% | 30.6 | 72.3% | 4.8 |
| | AQ_wat_dw2 | 10.2% | 50.6 | 17.6% | 10.3 | 9.7% | 18.8 | 25.7% | 15.4 | 36.9% | 6.7 |
| | AQ_wat_dw3 | 5.0% | 9.0 | 5.0% | 29.1 | 3.8% | 32.0 | 70.7% | 1.9 | 15.5% | 2.3 |
| 1 - biofilms | AQ_bio_dw1 | 8.6% | 23.5 | 25.0% | 19.1 | 3.9% | 74.1 | 56.7% | 6.9 | 5.8% | 7.9 |
| | AQ_bio_dw2 | 13.6% | 28.0 | 28.4% | 14.1 | 2.9% | 46.0 | 48.2% | 6.52 | 6.8% | 11.8 |
| | AQ_bio_dw3 | 3.4% | 17.1 | 13.9% | 18.4 | 5.5% | 39.3 | 72.1% | 1.85 | 5.2% | 29.8 |
| 2 - biofilms | AQ_bio_up1 | 32.2% | 14.5 | | | | | 61.3% | 9.5 | 6.8% | 23.7 |
| | AQ_bio_up2 | 56.6% | 12.6 | | | | | 36.4% | 18.3 | 7.0% | 15.7 |
| | AQ_bio_up3 | 44.0% | 6.6 | | | | | 48.1% | 8.1 | 7.8% | 10.8 |

* Two analyses are shown from sub-sampled datasets set at 1000 reads: (1) reads from WS, DB, IB, and aquifer waters from upstream the SIS were considered as the sources of taxa for the aquifer samples downstream the SIS; (2) reads from WS and the aquifer waters upstream the SIS were considered as the sources of taxa for the aquifer biofilms recovered upstream the SIS. SourceTracker was run 3 times using the 16S rRNA gene OTU contingency table and the default parameters. Relative contributions of the sources were averaged. Relative standard deviations (%RSD) are indicated, and used as confidence values. RSD > 100% indicates low confidence on the estimated value. WS: Watershed runoff waters; DB: Detention basin sediments; IB: Infiltration basin sediments. Sequences that could not be attributed to one of the tested sources were grouped under the term unknown.

[Figure]

**Figure S1. Boxplot representation of the 16S rRNA gene copy numbers measured by quantitative PCR per DNA extracts of runoff waters (WS), sediments from the detention basin (DB), soils from the infiltration basin (IB), aquifer waters (AQ_waters) or aquifer clay beads biofilms (AQ_bio).** Values were expressed per g of dry weight soil or sediment, or per mL, or per surface for the clay bead biofilms.

[Figure]

**Figure S2. Rarefaction curves showing the relation between the number of V5-V6 16S rRNA (*rrs*) gene reads analyzed and OTU numbers per compartment of the Mi-plaine watershed of Chassieu (France).** (a) without sub-sampling and (b) with a sub-sampling performed at 20,624 reads per sample.

Table S2. General features of the 16S rRNA and *tpm* genes meta-barcoding datasets per environmental sample used in this investigation according to Table S1, and their matching 16S rRNA gene copies per g dry weight of soil or sediment, or per ml or cm2 of clay beads.

| watershed compartment | description | sampling date | sample ID | 16S rRNA gene copies per g or ml or cm2 | | raw 16S rRNA gene reads | | | | cleaned 16S rRNA gene reads* | raw *tpm* gene reads | cleaned *tpm* gene reads** |
|---|---|---|---|---|---|---|---|---|---|---|---|---|
| | | | | | | total | length | | | total | total | total |
| | | | | mean | standard deviation | number | min | max | mean | number | number | number |
| WS | Runoff waters from Mi-plaine watershed | october 2014 | C1_1_2014o | 3,28E+08 | 9,40E+06 | 40927 | 38 | 515 | 417 | 10372 | 28 441 | 16347 |
| | | | C1_2_2014o | 3,43E+07 | 9,18E+05 | 147926 | 38 | 530 | 417 | 33655 | 31 450 | 16351 |
| | | | C1_3_2014o | 1,33E+09 | 1,51E+07 | 144070 | 37 | 564 | 416 | 31979 | 21 627 | 11945 |
| | | | C1_5_2014o | 1,73E+07 | 3,34E+06 | 164997 | 38 | 551 | 416 | 40110 | 18 394 | 8913 |
| | | | C1_6_2014o | 9,90E+08 | 4,88E+07 | 31933 | 38 | 510 | 416 | 9574 | 18 273 | 6923 |
| | | | C1_7_2014o | 7,49E+07 | 7,88E+06 | 182059 | 37 | 512 | 417 | 44190 | 26 650 | 13060 |
| | | | C1_8_2014o | not available | | 201635 | 38 | 556 | 416 | 44238 | 13 501 | 2688 |
| | | | C1_9_2014o | not available | | 176833 | 38 | 556 | 417 | 37588 | 28 330 | 15650 |
| | | | C1_10_2014o | 5,22E+07 | 0,00E+00 | 153508 | 35 | 564 | 417 | 37307 | 41 840 | 20887 |
| | | | C1_11_2014o | 8,44E+06 | 0,00E+00 | 159422 | 37 | 507 | 416 | 37141 | 40 421 | 20800 |
| | | | C1_13_2014o | not available | | 152100 | 38 | 507 | 417 | 36975 | 10 437 | 3866 |
| | | | C1_14_2014o | 1,24E+08 | 1,83E+06 | 189441 | 38 | 570 | 415 | 39401 | 29 486 | 19525 |
| | | | C1_15_2014o | 5,16E+08 | 4,12E+07 | 153174 | 37 | 548 | 417 | 37311 | 43 162 | 14283 |
| | | | C1_16_2014o | 1,55E+08 | 6,34E+06 | 116814 | 38 | 521 | 417 | 27140 | 36 019 | 18880 |
| | | | C1_17_2014o | 1,06E+08 | 2,62E+07 | 121294 | 37 | 537 | 417 | 28784 | 26 622 | 11424 |
| | | | C1_18_2014o | 6,23E+08 | 5,74E+06 | 25535 | 138 | 539 | 417 | 6507 | 22 557 | 8072 |
| | | | C1_19_2014o | 1,35E+08 | 1,39E+06 | 111031 | 38 | 539 | 417 | 26132 | 75 495 | 32920 |
| | | | C1_20_2014o | 1,84E+06 | 2,89E+05 | 111138 | 36 | 550 | 416 | 24198 | 68 389 | 11958 |
| | | | C1_21_2014o | 4,21E+05 | 0,00E+00 | 110366 | 38 | 533 | 416 | 23261 | 40 980 | 24656 |
| | | | C1_22_2014o | 7,71E+07 | 8,78E+06 | 102382 | 37 | 567 | 417 | 25138 | 42 066 | 18447 |
| | | march 2015 | C2_1_2015m | not available | | 236023 | 38 | 569 | 417 | 62451 | 264 | 74 |
| | | | C2_2_2015m | 1,50E+09 | 9,30E+07 | 224336 | 37 | 496 | 417 | 69288 | 33 025 | 9096 |
| | | | C2_3_2015m | 9,04E+08 | 3,57E+07 | 211508 | 37 | 546 | 418 | 63308 | 3 415 | 1010 |
| | | | C2_5_2015m | 5,78E+08 | 3,19E+07 | 199231 | 36 | 531 | 417 | 69205 | 64 | 1 |
| | | | C2_6_2015m | 9,73E+08 | 4,85E+07 | 233803 | 38 | 567 | 416 | 71619 | 44 248 | 14329 |
| | | | C2_7_2015m | 4,67E+08 | 4,58E+06 | 259717 | 38 | 558 | 417 | 85715 | 24 930 | 13150 |
| | | | C2_8_2015m | 2,21E+08 | 4,93E+07 | 217103 | 38 | 547 | 417 | 72654 | 37 906 | 20071 |
| | | | C2_9_2015m | 2,55E+08 | 7,89E+06 | 202232 | 38 | 539 | 417 | 62008 | 2 772 | 812 |
| | | | C2_10_2015m | 1,71E+09 | 1,25E+08 | 260742 | 37 | 525 | 417 | 86488 | 35 549 | 17955 |
| | | | C2_11_2015m | 5,01E+08 | 2,12E+07 | 292557 | 34 | 513 | 416 | 94733 | 27 495 | 13579 |
| | | | C2_13_2015m | 9,29E+07 | 3,59E+06 | 223565 | 37 | 552 | 417 | 74733 | 31 663 | 12951 |
| | | | C2_14_2015m | not available | | 390926 | 37 | 539 | 417 | 181207 | 6 382 | 2974 |
| | | | C2_15_2015m | 4,14E+08 | 3,76E+06 | 270504 | 38 | 493 | 416 | 90679 | 25 419 | 14455 |
| | | | C2_16_2015m | 5,65E+07 | 9,98E+06 | 201378 | 35 | 515 | 418 | 69440 | 44 649 | 19644 |
| | | | C2_17_2015m | 3,37E+08 | 1,23E+07 | 283468 | 38 | 475 | 417 | 88925 | 71 756 | 46251 |
| | | | C2_18_2015m | 7,48E+08 | 4,44E+07 | 270108 | 37 | 554 | 415 | 84620 | 36 915 | 25827 |
| | | | C2_19_2015m | 3,99E+08 | 1,13E+07 | 240674 | 38 | 513 | 417 | 77002 | 36 448 | 18540 |
| | | | C2_20_2015m | 3,01E+08 | 4,68E+06 | 248371 | 38 | 569 | 417 | 81746 | 28 239 | 15549 |
| | | | C2_21_2015m | 1,19E+08 | 7,48E+06 | 217320 | 38 | 536 | 418 | 79754 | 25 534 | 12414 |
| | | | C2_22_2015m | 4,83E+08 | 3,98E+07 | 191332 | 38 | 557 | 417 | 63976 | 32 456 | 17699 |
| | | september 2015 | C3_1_2015s | 1,28E+09 | 5,21E+07 | 217656 | 37 | 541 | 417 | 67274 | 45 066 | 11816 |
| | | | C3_2_2015s | 9,54E+08 | 4,51E+07 | 180442 | 38 | 562 | 417 | 52218 | 53 248 | 7328 |
| | | | C3_3_2015s | 3,50E+09 | 2,01E+08 | 222919 | 34 | 547 | 416 | 72826 | 28 050 | 8556 |
| | | | C3_5_2015s | 8,09E+08 | 3,76E+07 | 216319 | 38 | 549 | 417 | 66103 | 42 372 | 17737 |
| | | | C3_6_2015s | 5,43E+08 | 4,20E+07 | 187237 | 38 | 552 | 417 | 56021 | 93 620 | 27777 |
| | | | C3_7_2015s | 5,99E+08 | 3,34E+07 | 223766 | 37 | 541 | 418 | 81246 | 47 873 | 9264 |
| | | | C3_8_2015s | 7,97E+08 | 3,64E+07 | 214873 | 37 | 553 | 418 | 67870 | 86 760 | 20531 |
| | | | C3_9_2015s | 8,10E+08 | 3,88E+07 | 188331 | 34 | 550 | 417 | 57162 | 52 682 | 20656 |
| | | | C3_10_2015s | 1,37E+08 | 3,97E+06 | 133734 | 38 | 550 | 418 | 42353 | 121 502 | 32924 |
| | | | C3_11_2015s | 2,40E+09 | 5,16E+07 | 226106 | 37 | 533 | 418 | 70286 | 48 775 | 26844 |
| | | | C3_13_2015s | 5,92E+08 | 1,14E+07 | 163771 | 38 | 543 | 418 | 43970 | 30 810 | 1892 |
| | | | C3_14_2015s | 5,24E+08 | 4,47E+07 | 242705 | 38 | 497 | 417 | 68298 | 37 299 | 25105 |
| | | | C3_15_2015s | 2,39E+08 | 1,51E+07 | 164798 | 38 | 556 | 418 | 53480 | 70 582 | 21685 |
| | | | C3_16_2015s | 1,66E+08 | 8,73E+06 | 206307 | 36 | 535 | 417 | 68792 | 33 571 | 10263 |
| | | | C3_17_2015s | 7,02E+08 | 4,33E+07 | 273162 | 36 | 552 | 416 | 82415 | 58 208 | 33772 |
| | | | C3_18_2015s | 7,02E+08 | 4,96E+07 | 228375 | 38 | 550 | 417 | 64981 | 45 722 | 16988 |
| | | | C3_19_2015s | 3,55E+08 | 2,47E+07 | 277908 | 33 | 555 | 417 | 93895 | 43 829 | 16895 |
| | | | C3_20_2015s | 1,21E+09 | 7,79E+07 | 245342 | 34 | 553 | 417 | 81970 | 47 399 | 17312 |
| | | | C3_21_2015s | 1,92E+09 | 1,55E+08 | 268179 | 36 | 554 | 416 | 83043 | 41 323 | 20059 |
| | | | C3_22_2015s | 8,49E+08 | 6,54E+07 | 222664 | 35 | 554 | 417 | 61898 | 35 005 | 16236 |
| DB | Sediment deposits from the detention basin (see Marti et al., 2017) | october 2013 | BR_2013o_P1 | 4,96E+10 | 6,68E+08 | 103468 | 44 | 539 | 416 | 21749 | 53 711 | 12406 |
| | | | BR_2013o_P2 | 2,49E+11 | 2,87E+10 | 113868 | 65 | 539 | 419 | 21120 | 43376 | 21073 |
| | | | BR_2013o_P4 | 4,18E+11 | 2,66E+10 | 139174 | 38 | 539 | 416 | 30946 | 45255 | 10747 |
| | | | BR_2013o_P7 | 1,43E+11 | 9,47E+09 | 126713 | 38 | 539 | 416 | 28790 | 25108 | 8973 |
| | | april 2014 | BR_2014a_P1 | 2,90E+11 | 1,73E+10 | 119627 | 34 | 546 | 417 | 24591 | 37898 | 14470 |
| | | | BR_2014a_P2 | 3,81E+11 | 1,18E+10 | 118079 | 37 | 538 | 418 | 23373 | 51237 | 20635 |
| | | | BR_2014a_P4 | 7,29E+11 | 2,00E+10 | 121050 | 38 | 548 | 417 | 24456 | 75858 | 29945 |
| | | | BR_2014a_P7 | 8,16E+10 | 3,76E+09 | 111719 | 37 | 555 | 417 | 22155 | 37057 | 5108 |
| | | february 2014 | BR_2014f_P1 | 1,56E+10 | 1,54E+09 | 135391 | 34 | 539 | 416 | 27325 | 55677 | 16387 |
| | | | BR_2014f_P2 | 3,51E+11 | 2,41E+10 | 137484 | 38 | 556 | 415 | 28735 | 46872 | 19983 |
| | | | BR_2014f_P4 | 2,27E+11 | 1,98E+10 | 19308 | 80 | 504 | 415 | 6062 | 44648 | 5313 |
| | | | BR_2014f_P7 | 2,44E+11 | 2,03E+10 | 115571 | 38 | 539 | 416 | 25348 | 38184 | 8690 |
| | | july 2014 | BR_2014j_P1 | 2,98E+08 | 1,38E+08 | 125238 | 38 | 540 | 419 | 21026 | 38261 | 12278 |
| | | | BR_2014j_P2 | 2,07E+11 | 8,73E+08 | 113171 | 38 | 541 | 419 | 20624 | 40765 | 17632 |
| | | | BR_2014j_P4 | 1,91E+11 | 9,99E+09 | 159988 | 38 | 552 | 417 | 32206 | 58429 | 20688 |
| | | | BR_2014j_P7 | 1,76E+11 | 3,63E+09 | 124702 | 33 | 551 | 419 | 22131 | 39442 | 9670 |
| | | april 2015 | BR_2015a_P1 | 2,93E+10 | 1,48E+08 | 210960 | 37 | 546 | 418 | 61781 | 13 448 | 4735 |
| | | | BR_2015a_P2 | 2,11E+10 | 3,31E+09 | 185283 | 38 | 554 | 419 | 49779 | 21 165 | 6612 |
| | | | BR_2015a_P4 | 8,88E+09 | 3,47E+08 | 193051 | 36 | 539 | 418 | 57523 | 15 313 | 4636 |
| | | | BR_2015a_P7 | 8,80E+09 | 1,33E+09 | 254539 | 33 | 539 | 417 | 74225 | 46 713 | 4674 |
| IB | Sediments samples (0-50 cm depth) from the infiltration basin | november 2015 | VF1_ZA_2015n | 1,37E+11 | 2,54E+10 | 212619 | 38 | 556 | 417 | 59766 | 70 675 | 24919 |
| | | | VF2_ZA_2015n | 1,21E+11 | 1,11E+10 | 187624 | 36 | 556 | 419 | 52586 | 29 417 | 13441 |
| | | | VF3_ZA_2015n | 1,46E+11 | 1,38E+10 | 203861 | 38 | 556 | 418 | 51732 | 37 168 | 12663 |
| | | | VF4_ZA_2015n | 1,05E+11 | 3,63E+10 | 181177 | 37 | 533 | 419 | 48886 | 30 174 | 12410 |
| | | | VF5_ZA_2015n | 9,45E+10 | 1,24E+10 | 173629 | 35 | 556 | 418 | 46316 | 28 717 | 11974 |
| | | | VF6_ZA_2015n | 1,04E+11 | 1,31E+10 | 204025 | 35 | 557 | 418 | 55523 | 11 586 | 3294 |
| | | | VF7_ZB_2015n | 8,73E+10 | 2,94E+10 | 193858 | 36 | 558 | 418 | 54171 | 20 525 | 6405 |
| | | | VF8_ZB_2015n | 1,02E+11 | 4,65E+10 | 198776 | 38 | 556 | 419 | 48193 | 15 654 | 5370 |
| | | | VF9_ZB_2015n | 3,46E+11 | 3,16E+11 | 116411 | 38 | 512 | 417 | 33815 | 68 510 | 20002 |
| | | | VF10_ZB_2015n | 1,18E+11 | 1,44E+10 | 206561 | 33 | 553 | 419 | 52587 | 40 508 | 15119 |
| | | | VF11_ZH_2015n | 1,32E+11 | 7,56E+10 | 225377 | 38 | 555 | 418 | 56011 | 52 829 | 16520 |
| | | | VF12_ZH_2015n | 1,14E+11 | 6,10E+10 | 221276 | 38 | 555 | 418 | 55269 | 49 024 | 14936 |
| | | | VF13_ZH_2015n | 1,32E+11 | 3,20E+10 | 203512 | 34 | 556 | 419 | 50839 | 47 849 | 19000 |
| | | | VF14_ZH_2015n | 1,04E+11 | 2,55E+10 | 70712 | 38 | 493 | 417 | 29133 | 20 644 | 3728 |
| | | | VF15_ZH_2015n | 1,42E+11 | 4,42E+10 | 211533 | 37 | 558 | 419 | 57315 | 25 876 | 8327 |
| AQ_bio_up | Aquifer biofilm sample | september 2015 | JBio_Am1_2015s | 6,63E+06 | 1,95E+05 | 57665 | 29 | 513 | 409 | 25809 | 79 053 | 43527 |
| | | | JBio_Am2_2015s | 7,03E+06 | 1,03E+06 | 155688 | 26 | 513 | 409 | 61754 | 16 246 | 13548 |
| | | | JBio_Am3_2015s | 6,16E+06 | 6,05E+05 | 122322 | 28 | 546 | 409 | 51420 | 55 679 | 38935 |
| AQ_bio_dw | Aquifer biofilm sample | | JBio_Av1_2015s | 1,92E+07 | 1,24E+06 | 118783 | 28 | 521 | 408 | 39167 | 103 521 | 77805 |
| | | | JBio_Av2_2015s | 1,35E+07 | 7,53E+05 | 109023 | 29 | 516 | 408 | 32445 | 55 807 | 43244 |
| | | | JBio_Av3_2015s | 3,06E+08 | 5,07E+07 | 125969 | 29 | 553 | 409 | 38774 | 31 379 | 23381 |
| AQ_wat_up | Aquifer water sample | | JEau_AmJ0_2015s | 7,17E+03 | 5,81E+02 | 154982 | 29 | 507 | 410 | 59529 | 51 633 | 41145 |
| | | | JEau_AmJ10_2015s | 6,94E+03 | 5,13E+02 | 115472 | 29 | 546 | 411 | 32476 | 76 791 | 48320 |
| | | | JEau_AmJ5_2015s | 6,86E+03 | 1,65E+02 | 169773 | 31 | 537 | 410 | 64731 | 72 452 | 50813 |
| AQ_wat_dw | Aquifer water sample | | JEau_AvJ0_2015s | 3,55E+04 | 1,53E+03 | 107788 | 40 | 529 | 409 | 31124 | 19 782 | 4718 |
| | | | JEau_AvJ10_2015s | 9,21E+03 | 1,23E+03 | 176475 | 29 | 544 | 409 | 42988 | 29 686 | 21072 |
| | | | JEau_AvJ5_2015s | 1,61E+04 | 1,10E+03 | 90793 | 29 | 514 | 409 | 31963 | 49 721 | 33735 |

*: Mean *rrs* read length was 408 bp, maximum 415 bp, and minimum 375 bp.

**: Mean *tpm* read length was 215 bp, maximum 233 bp, and minimum 195 bp.

WS: Watershed runoff waters; DB: Detention basin sediments IB: Infiltration basin sediments ; AQ_wat: Aquifer waters; AQ_bio: Aquifer biofilms.

Table S3. Number of *tpm* reads among blank samples run during the tpm meta-barcoding procedure, and their taxonomic allocation and relatedness to OTUs recovered from the environmental samples. *: restricted to above ground samples; **: not considered in the coalescence analysis, see Table S8.

| blank sample OTU | total number of reads | identical OTU sequence among the environmental samples | maximum % identity with environmental *tpm* sequences | genus | species | blank 1 (soil) | blank 2 (soil) | blank 3 (water) | blank 4 (water) | blank 5 (water) |
|---|---|---|---|---|---|---|---|---|---|---|
| Otu01 | 867 | Otu00573* | 100 | Pseudomonas | unclassified | 0 | 0 | 0 | 867 | 0 |
| Otu02 | 118 | | 99 | Pseudomonas | fluorescens | 0 | 0 | 0 | 118 | 0 |
| Otu03 | 21 | | 99 | Pseudomonas | fluorescens | 21 | 0 | 0 | 0 | 0 |
| Otu04 | 17 | | no match | unclassified | unclassified | 1 | 0 | 15 | 0 | 0 |
| Otu05 | 17 | Otu00151* | 100 | Pseudomonas | xanthomarina | 0 | 0 | 8 | 9 | 0 |
| Otu06 | 13 | | no match | unclassified | unclassified | 1 | 0 | 12 | 0 | 0 |
| Otu07 | 10 | | 99 | Pseudomonas | unclassified | 0 | 0 | 0 | 10 | 0 |
| Otu08 | 7 | | no match | unclassified | unclassified | 0 | 1 | 6 | 0 | 0 |
| Otu09 | 7 | | 99 | Pseudomonas | unclassified | 0 | 0 | 0 | 7 | 0 |
| Otu10 | 6 | | no match | unclassified | unclassified | 1 | 0 | 5 | 0 | 0 |
| Otu11 | 5 | | no match | unclassified | unclassified | 0 | 0 | 5 | 0 | 0 |
| Otu12 | 4 | Otu01054** | 100 | unclassified | unclassified | 0 | 0 | 0 | 3 | 0 |
| Otu13 | 3 | | 99 | Pseudomonas | unclassified | 0 | 0 | 0 | 3 | 0 |
| Otu14 | 3 | Otu00069 | 100 | Pseudomonas | fragi | 0 | 0 | 3 | 0 | 0 |
| Otu15 | 3 | Otu00002* | 100 | Pseudomonas | unclassified | 0 | 0 | 2 | 1 | 0 |
| Otu16 | 2 | | 99 | Pseudomonas | unclassified | 0 | 0 | 0 | 2 | 0 |
| Otu17 | 2 | | no match | unclassified | unclassified | 0 | 0 | 0 | 1 | 0 |
| Otu18 | 2 | | 98 | unclassified | unclassified | 0 | 0 | 2 | 0 | 0 |
| Otu19 | 2 | Otu00519** | 100 | unclassified | unclassified | 0 | 0 | 0 | 1 | 1 |
| Otu20 | 2 | | 99 | Pseudomonas | unclassified | 0 | 0 | 0 | 2 | 0 |
| Otu21 | 2 | | 99 | Pseudomonas | unclassified | 0 | 0 | 0 | 2 | 0 |
| Otu22 | 2 | Otu00556** | 100 | unclassified | unclassified | 0 | 2 | 0 | 0 | 0 |
| Otu23 | 2 | | 99 | Pseudomonas | unclassified | 0 | 0 | 0 | 2 | 0 |

---

## Referee Report (RR1)

Specific comments

Please write the units always in the same way (for example, x y$^{-1}$) throughout the manuscript.

L153. "0.4 mg mL-1" should be "0.4 mg mL$^{-1}$"

L155. Remove the comma after "-80°C"

---

## Referee Report (RR2)

The authors of "Coalescence of bacterial groups originating from urban runoffs and artificial infiltration systems among aquifer microbiomes" Colin et al., carefully considered and incorporated the critical feedback by both reviewers into the edited manuscript (displayed through their curated point-by-point response). Notably, the detailed materials and methods will assist future readers in interpretation, replicability, and placing the methods into a field in a conversation over standard operating procedures. Also, I encourage all future readers of the manuscript to engage with the supplemental materials in a considered and careful manner because of the rich and robust data that is supporting the reported findings in the document.

This subsequent read through and second round of comments produced only minor copy-editing level suggested changes listed below. Line numbers refer to the final document, not the track changes version.

Please standardize the spacing between the unit and the number throughout the text (e.g., L92 vs. L113, although different units, suggest maintaining the space throughout).

L18 – Please remove the hyphen in "micro-organisms" here and throughout the text.

L49 – Please provide a comma after "i.e." here and throughout the text.

L116 – "Urbanistic changes" presents a challenge to me in readability. Perhaps simply "modifications" would suffice.

L145 – Please replace "About" with "Approximately"

L146 – Please alter "Total DNAs were" to "Total DNA was". Throughout the text, convert the usage of DNAs plural into DNA singular and alter the referencing verbs accordingly.

L149 – Thank you for detailing all of the carefully considered controls here and throughout the text.

L208 – Here, and throughout, please ensure that a decimal point rather than decimal comma is used in all numbers (0,07 to 0.07).

L381 – Please add a comma before "but"

L547 – Please remove the comma before "and"

Additionally, thank you for the robust supplemental materials. I believe these will enable the incorporation of the presented study into future investigations of subsurface community assemblages.

---

## Author Response (AR2)

Dear HESS Editor,                                                    21/07/2020

Please find below a point-by-point response (rep) to the second round of reviews (R1b and R2b), including a description of all relevant changes made in the manuscript, and a marked-up manuscript version showing in red modifications performed on the text.

All issues raised by the reviewers were considered. These modifications were grammatical or aimed at improving technical terms.

Thank you for the time spent evaluating our article,

Best regards,
B. Cournoyer
* * *
Referee #1: Carles, Louis  louis.carles@eawag.ch

Second review - comments :
R1b-1. Please write the units always in the same way (for example, x y-1) throughout the manuscript. L153. "0.4 mg mL-1" should be "0.4 mg mL-1"

**Rep-R1b-1:** modifications were made accordingly all over the manuscript

R1b-2. L155. Remove the comma after "-80°C"

**Rep-R1b-2:** the comma was removed
* * *
Referee #2: Anonymous

Second review - comments :

R2b-1. Please standardize the spacing between the unit and the number throughout the text (e.g., L92 vs. L113, although different units, suggest maintaining the space throughout).

**Rep-R2b-1:** fixed accordingly all over the manuscript

R2b-2. L18 – Please remove the hyphen in "micro-organisms" here and throughout the text.

**Rep-R2b-2:** fixed accordingly all over the manuscript

R2b-3. L49 – Please provide a comma after "i.e." here and throughout the text.

**Rep-R2b-3:** fixed accordingly all over the manuscript

R2b-4. L116 – "Urbanistic changes" presents a challenge to me in readability. Perhaps simply "modifications" would suffice.

**Rep-R2b-4:** fixed as follow, "No significant modifications impacting the urban watershed were recorded during the investigation."

R2b-5. L145 – Please replace "About" with "Approximately"

**Rep-R2b-5:** modified accordingly

R2b-6. L146 – Please alter "Total DNAs were" to "Total DNA was". Throughout the text, convert the usage of DNAs plural into DNA singular and alter the referencing verbs accordingly.

**Rep-R2b-6:** fixed accordingly all over the manuscript

R2b-7. L208 – Here, and throughout, please ensure that a decimal point rather than decimal comma is used in all numbers (0,07 to 0.07).

**Rep-R2b-7:** fixed accordingly all over the manuscript

R2b-8. L381 – Please add a comma before "but"

**Rep-R2b-8:** modified accordingly

R2b-9. L547 – Please remove the comma before "and"

**Rep-R2b-9:** modified accordingly

[revised manuscript text omitted]